# Lifts of Symmetric Tensors: Fluids, Plasma, and Grad Hierarchy

**DOI:** 10.3390/e21090907

**Published:** 2019-09-18

**Authors:** Oğul Esen, Miroslav Grmela, Hasan Gümral, Michal Pavelka

**Affiliations:** 1Department of Mathematics, Gebze Technical University, 41400 Gebze-Kocaeli, Turkey; 2École Polytechnique de Montréal, C.P.6079 suc. Centre-ville, Montréal, QC H3C 3A7, Canada; miroslav.grmela@polymtl.ca; 3Department of Mathematics, Yeditepe University Atasehir, 34755 Istanbul, Turkey; hasangumral@gmail.com; 4Mathematical Institute, Faculty of Mathematics and Physics, Charles University, Sokolovská 83, 186 75 Prague, Czech Republic

**Keywords:** diffeomorphisms group, cotangent lift, vertical representative, fluids, kinetic theory, entropy

## Abstract

Geometrical and algebraic aspects of the Hamiltonian realizations of the Euler’s fluid and the Vlasov’s plasma are investigated. A purely geometric pathway (involving complete lifts and vertical representatives) is proposed, which establishes a link from particle motion to evolution of the field variables. This pathway is free from Poisson brackets and Hamiltonian functionals. Momentum realizations (sections on $T^{*}T^{*}Q$) of (both compressible and incompressible) Euler’s fluid and Vlasov’s plasma are derived. Poisson mappings relating the momentum realizations with the usual field equations are constructed as duals of injective Lie algebra homomorphisms. The geometric pathway is then used to construct the evolution equations for 10-moments kinetic theory. This way the entire Grad hierarchy (including entropic fields) can be constructed in a purely geometric way. This geometric way is an alternative to the usual Hamiltonian approach to mechanics based on Poisson brackets.


At ubi materia, ibi Geometria.*(Johannes Kepler)*


## 1. Introduction

A physical system with state variables *x* is said to be in the Hamiltonian form if there exists a Poisson bracket {•,•} and a Hamiltonian function(al) H, such that the equation of motion governing the dynamics can be written as(1)$$\dot{x}=\left\{x,H\right\}.$$

Mechanics can be typically cast into a Hamiltonian form; the state variables can be for instance particle position and momentum, rigid body angular momentum [1], distribution functions in kinetic theory [2,3], hydrodynamic fields [4,5], electromagnetic fields [6], etc. An advantage of the geometric formulation is that it provides additional leads towards proper coupling of the particular theories, e.g., MHD [7], as well as automatic consistency with mechanics. Such properties can be used also in the numerical simulations [8,9,10,11,12]. As a manifestation of the skew symmetry of the Poisson bracket, the Hamiltonian function H is conserved throughout the motion. That means that Hamiltonian systems are energy-preserving, which manifests the reversible character of the Hamiltonian dynamics, see also [13] for discussion of time-reversal symmetry and Onsager–Casimir reciprocal relations. It is possible to generalize the Hamiltonian framework in a way that is proper also for irreversible systems. This generalization is called GENERIC in the literature [13,14,15,16,17]. In this present work we shall stay in the area of Hamiltonian systems so that in the reversible framework.

This generalization has started to appear in works by the authors of [18,19,20,21]. The combination of Hamiltonian and gradient dynamics has been called metriplectic in the works by the authors of [22,23] and GENERIC (General Equation for Nonequilibrium Reversible–Irreversible Coupling) in the works by the authors of [13,14,15,17] where the Riemannian (metric or its nonlinear generalization) geometrical structure appearing in the gradient (dissipative) part of the vector field is more general. The search for the most appropriate, both from the physical and the mathematical point of view, formulation of the combination of the Hamiltonian and the gradient dynamics continues. For example, contact geometry offers a very convenient setting [13,24,25]. A variational formulation has recently been introduced in the works by the authors of [26,27]. Another variational formulation also arises in the contact geometry formulation [28].

While performing the Hamiltonian analysis of a given differential system, two tasks, namely, determining a Poisson bracket and choosing a Hamiltonian function, must be achieved simultaneously. It is indeed a hard task to arrive at a Hamiltonian realization of an arbitrary system. Cotangent bundle, which can physically be considered as the momentum phase space of a configuration manifold, carries a canonical symplectic two-form, thus a canonical Poisson structure. This canonical geometry fits well many physical systems involving the classical mechanics and the electromagnetic theory [2,29]. If the physical system possesses some constraints or/and some symmetries, then such a canonical Poisson framework cannot be proper. In this case, a reduction procedure should be applied in order to arrive at a correct Poisson picture [30].

**Lie–Poisson Equations**. For many physical systems, the configuration space is a Lie group *G*, see the work by the authors of [31]. In this case, the group action induces a symmetry on the cotangent bundle $T^{*}G$ of the Lie group, so that the canonical symplectic structure on $T^{*}G$ reduces to a Poisson structure on the quotient space $T^{*}G/G$. This reduction procedure has rich geometric and algebraic properties. Let us depict this more explicitly. Consider a Lie group *G* and its Lie algebra g equipped with a Lie bracket ${\left[•,•\right]}_{g}$. Linear algebraic dual $g^{*}$ of the Lie algebra carries a natural Poisson structure called as the Lie–Poisson bracket [30,32,33,34]. Under the reflexivity condition, the Lie–Poisson bracket for $\mu \in g^{*}$ is given by,(2)$${\left\{F,H\right\}}_{g^{*}}\left(\mu \right)=\pm \langle \mu ,{\left[\frac{\delta F}{\delta \mu },\frac{\delta H}{\delta \mu }\right]}_{g}\rangle ,$$
where F and H are two function(al)s defined on the dual space $g^{*}$, and the pairing at the right hand side is the one between $g^{*}$ and g. The bracket inside the pairing in Equation (Equation 2) is the Lie algebra bracket on g, and the notation δH/δμ stands for the Fréchet derivative of the functional H. In this case, the dynamics is governed by the Lie–Poisson equations(3)$$\dot{\mu }={\left\{\mu ,H\right\}}_{g^{*}}=\mp ad_{\frac{\delta H}{\delta \mu }}^{*}\Pi ,$$
where $ad^{*}$ denotes the coadjoint representation of g on $g^{*}$, obtained by the dualization of the adjoint action of g on itself. Here, the adjoint action is defined to be the Lie algebra bracket on g.

**Diffeomorphism Groups**. The configuration spaces of some of the physical systems, such as fluids and plasma theories, are diffeomorphism groups [31,35,36,37]. Diffeomorphism groups are infinite dimensional Lie groups [38,39,40]. In these cases, the Lie–Poisson formulation, presented in the previous paragraph, takes the following particular form. Assume that a continuum rests in a manifold M in $R^{3}$, then diffeomorphism group $Diff\left(M\right)$ acts from left on the particle space M by evaluation, whereas right action commutes with the particle motion and constitutes an infinite dimensional symmetry group of the kinematic description. This is called the particle relabeling symmetry [41]. We assume the Lie algebra of $Diff\left(M\right)$ as the space $X\left(M\right)$ of smooth vector fields. Here, the Lie algebra bracket is the minus of the Jacobi–Lie bracket of vector fields, that is,(4)$${\left[X,Y\right]}_{X\left(M\right)}=-{\left[X,Y\right]}_{JL}=-L_{X}Y,$$
where $L_{X}$ denotes the Lie derivative operator. We define the dual space $X^{*}\left(M\right)$ of the Lie algebra as the space of one-form densities $\Lambda^{1}\left(M\right)⊗Den\left(M\right)$ on M. Here, the pairing between a vector field *X* and a dual element Π⊗dμ is defined as(5)$$\langle \Pi ⊗d\mu ,X\rangle =\int_{M}\Pi \left(z\right)\cdot X\left(z\right)\;d\mu \left(z\right),$$
where the pairing inside the integral is the canonical one between the covector Π(z) and the vector X(z). Here, dμ is a density, that is, a volume form on M. The adjoint action of the Lie algebra onto itself is defined by the Lie algebra bracket in (Equation 4). A simple calculation(6)$$-\langle ad_{X}^{*}\left(\Pi ⊗d\mu \right),Y\rangle =\langle \Pi ⊗d\mu ,ad_{X}Y\rangle \;\mathrm{with}\;ad_{X}Y=L_{X}Y=\left[X,Y\right]$$
shows that the coadjoint action of the Lie algebra $X\left(M\right)$ on its dual $X^{*}\left(M\right)$ is(7)$$ad_{X}^{*}\left(\Pi ⊗d\mu \right)=\left(L_{X}\Pi +\left(div_{d\mu }X\right)\Pi \right)⊗d\mu ,$$
where $div_{d\mu }X$ denotes the divergence of the vector field *X* with respect to the volume form dμ. At this point, without lost of generalization, we fix the volume form dμ, so that we particularly consider a dual element as a one-form Π. For this choice, a Hamiltonain functional H generates the Lie–Poisson equations on the dual space $X^{*}\left(M\right)$, given by(8)$$\dot{\Pi }=-L_{\partial H/\partial \Pi }\Pi -\left(div_{d\mu }\frac{\partial H}{\partial \Pi }\right)\Pi .$$

In the divergence-free case (corresponding to volume-preserving motion), where the second term on the right hand side of Equation (Equation 8) identically vanishes, we obtain equations(9)$$\dot{\Pi }=-L_{X}\Pi .$$

**A geometric pathway to the dynamics of the continuum**. To describe the motion of a continuum, one may start to write down the whole microscopic data, involving the interactions, which is very difficult. The kinetic theory uses statistical concepts to handle practical problems of microscopic theory. In previous publications [42,43], a geometric pathway has been proposed for the incompressible Lie–Poisson Equations (Equation 9). Even though we present this geometry in the upcoming section, let us briefly summarize this geometrization. In order to arrive at the Lie–Poisson dynamics, start with the most basic ingredient of the theory, a vector field *X* generating the motion of a single particle, then, by applying purely geometric operations such as complete cotangent lifts and vertical representatives, define a generalized vector field $VX^{c*}$ on the space of sections. This results with the Lie–Poisson dynamics presented in (Equation 9). Explicitly, it has been proved in the works by the authors of [42,43] that the Lie–Poisson equations can be written in the form of(10)$${\dot{\Pi }}^{v}=VX^{c*}\left(x,\Pi \right),$$
where $VX^{c*}$ is the vertical (evolutionary), representative of the complete cotangent lift $X^{c*}$ of the vector field *X* generating particle motion. Here, ${\dot{\Pi }}^{v}$ is a vector field defined as the vertical lift of the one-form $\dot{\Pi }$, see, for example, the work by the authors of [44]. Note that, the geometrization procedure in Equation (Equation 10) for the Lie–Poisson Equations (Equation 9) is free of a Hamiltonian functional, and thus free of a Poisson structure. As we shall argue in the main body of the paper we believe that such a geometrization looks promising for continuum theories.

Let us make a mathematical remark here to depict this geometrization in more technical terms. A single particle traces a curve in the configuration space, so that its motion is determined by an ordinary differential equation (ODE) with time as the independent variable. On the other hand, the motion of the whole continuum is determined by a partial differential equation (PDE) governing a (scalar or vectorial) field. That is, mathematically, the single particle motion is a submanifold of a tangent bundle, whereas the motion of the continuum is a submanifold a jet bundle. Therefore, in order to find a link between these two motions, one needs to propose a geometric operator taking a tangent vector to a jet bundle element. Further, the final product of this operator must forget the motion on the base level in order just to concentrate on the field parameters. The geometrization in Equation (Equation 10) does both of these two tasks simultaneously, by relating the base motion to the motion of the field Π as well as by removing affects of the dynamics on the base level.

**The goal of the present paper**. In the literature, the geometrization (Equation 10) has only been studied for volume preserving dynamics [42,43], and no relationship with compressible dynamics has been established so far. Our aim in the present work is to extend the application area of the geometric pathway (Equation 10) to compressible systems and establish purely geometrical relations between all possible realizations (momentum or the usual field equations). The geometrization (Equation 10) depends on the duality between vector and covector fields, our first novel result is to put the averaged 2D-Euler equation into this framework by redefining the duality by a Sobolev norm. For further generalization, we define an injective Lie algebra homomorphism called Generalized Complete Cotangent Lift (GCCL). GCCL is a mapping from the space of symmetric tensors on a manifold to Hamiltonian vector fields on the cotangent bundle of the manifold. The dual of GCCL is a Poisson mapping relating the Vlasov dynamics in the momentum formulation to (both compressible and incompressible) Euler’s fluids in the momentum formulation. Then, using GCCL, compressible fluid motion is put into the framework of (Equation 10). As a byproduct, a unique decomposition of the space of symmetric tensors is established as a matched pair Lie algebra. We further equip the compressible fluid flow with entropy. Then, we study 10-moment hierarchy, which paves the way towards to whole Grad hierarchy including entropic moments. In summary, we provide an alternative purely geometric construction of mechanics in kinetic theory and Grad hierarchy.

Let us sum up the main novelty of this work. The Lie–Poisson dynamics is determined by the Lie–Poisson bracket and a Hamiltonian function(al), see, for example, the work by the authors of [30]. In the present work, we first establish that (see Section 2.3) one can write the Lie–Poisson dynamics for incompressible fluid and Vlasov plasma without referring to a Poisson bracket or a Hamiltonian function(al). Inspired by a series of papers on the moments of the Vlasov dynamics, see, for example, the works by the authors of [45,46,47,48,49], we propose a Lie algebra homomorphism GCCL from the symmetric contravariant tensors on a manifold to the Hamiltonian vector fields on its cotangent bundle (see Section 3.2). Dualization of this mapping determines the moments of the Lie–Poisson equations of form (Equation 9) (see Section 3.5). Double application of the GCCL operator and geometric pathway (Equation 10) lead to the compressible fluid dynamics (see Section 4.1) and the 10-moment approximation (see Section 4.2) free from a Poisson bracket and a Hamiltonian function(al).

**The contents**. This paper comprises three main sections. In the following section, we present the geometrical and algebraic foundations of the geometrization (Equation 10) as well as comment on underlying physical intuitions. Examples of this geometry are incompressible fluid flow and Vlasov’s plasma. We close this section by redefining the duality with a Sobolev metric and writing an averaged 2D-Euler’s equation in the form of (Equation 10). In the third section, Lie algebra of symmetric contravariant tensor fields and a matched pair decomposition of this space is established. An injective Lie algebra homomorphism, which we call Generalized Complete Cotangent Lift (GCCL), is defined. It is shown that the dual of GCCL is a Poisson mapping from the plasma level to the fluid level both in momentum formulations. In the fourth section, we generalize the geometrization (Equation 10) to compressible fluid motion including entropy density. Finally, the procedure is generalized to also cover the 10-moment Grad hierarchy in kinetic theory and incorporate a complimentary hierarchy of kinetic moments.

**Notation.** Let M and Q be finite dimensional manifolds equipped with the local coordinates $\left(x^{a}\right)$ and $\left(q^{i}\right)$, respectively. Cotangent bundles $T^{*}M$ and $T^{*}Q$ carry canonical symplectic two-forms, in Darboux’ coordinates given by $\Omega_{M}=dx^{a}\wedge d\Pi_{a}$ and $\Omega_{Q}=dq^{i}\wedge dp_{i}$, respectively. The Hamilton’s equation is generated by a Hamiltonian function *H* on $T^{*}Q$, and it is defined as(11)$$\iota_{X_{h}}\Omega_{Q}=dH,$$
where ι is the contraction operator. Further, it is assumed that all the requirements of functional analytic issues, such as existence, uniqueness, regularity, and convergence, are satisfied.

## 2. A Geometric Pathway to Kinetic Theories

### 2.1. Complete Tangent and Cotangent Lifts

Let M be an m−dimensional manifold equipped with local coordinates $\left(x^{a}\right)$. A vector field *X* on M generates a flow on the manifold M, say $\phi_{t}:M\to M$, which describes motion on the manifold with respect to a parameter *t* (representing the time). Locally, this reads that the vector field $X\left(x\right)=X^{a}\left(x\right)\partial /\partial x^{a}$ generates infinitesimal transformations:(12)$$x^{a}\to {\bar{x}}^{a}=x^{a}+tX\left(x^{a}\right)=x^{a}+tX^{b}\frac{\partial x^{a}}{\partial x^{b}}=x^{a}+tX^{a}.$$

#### 2.1.1. Complete Tangent Lift

Let us first recall the concept of complete cotangent lift, see, e.g., the work by the authors of [50], including its physical interpretation for clarity. Attaching a tangent plane to each point of the manifold, i.e., constructing the tangent bundle TM with induced local coordinates $\left(x^{a},v^{a}\right)$, the vector field *X* tells which vector of the tangent plane at each point describes the motion. In other words, the vector field tells the direction and the velocity with which the motion continues from the point, and it can be interpreted as a mapping from M to TM. To see this, consider the flow $\phi_{t}^{c}$ on the tangent bundle TM defined by the following equation.(13)$$\tau_{M}\circ \phi_{t}^{c}=\phi_{t}\circ \tau_{M},$$
where $\tau_{M}$ is the tangent bundle projection mapping a vector in TM to its initial point in M. Flow $\phi_{t}^{c}$ constitutes a one-parameter group of diffeomorphisms on TM called the complete tangent lift of the flow. From the differentiation of Equation (Equation 13) with respect to *t* at t=0 we define the complete tangent lift $X^{c}$ of *X* as follows(14)$$T\tau_{M}\circ X^{c}=X\circ \tau_{M}.$$

Notice that $X^{c}$ is a vector field on TM. Consider now a vector $v\in T_{x}M$, which by definition corresponds to a curve γ(t) passing through *x* satisfying $\dot{\gamma }=v$. The transformation (Equation 12) maps the curve to a new curve $\bar{x}\left(\gamma \left(t\right)\right)$. Therefore, the transformation maps the vector *v* to a new tangent vector with components(15)$${\bar{v}}^{b}=\frac{d}{dt}{\bar{x}}^{b}\left(\gamma \left(t\right)\right)=\frac{\partial {\bar{x}}^{b}}{\partial x^{a}}{\dot{\gamma }}^{a}\left(t\right)=\left(\delta_{a}^{b}+t\frac{\partial X^{b}}{\partial x^{a}}\right)v^{a}=v^{b}+tv^{a}\frac{\partial X^{b}}{\partial x^{a}}.$$

From the perspective of the tangent bundle TM, a vector field *X* induces both motion in the $\partial /\partial x^{a}$ direction as well as motion in the $\partial /\partial v^{b}$ direction, that is,(16)$$X^{c}\left(x,v\right)=X^{a}\frac{\partial }{\partial x^{a}}+v^{b}\frac{\partial X^{a}}{\partial x^{b}}\frac{\partial }{\partial v^{a}}.$$

#### 2.1.2. Complete Cotangent Lift

Consider the cotangent bundle $T^{*}M$ equipped with conjugate coordinates $\left(x^{a},\Pi_{a}\right)$. The complete cotangent lift of a flow $\varphi_{t}$ (of a vector field *X*) on M is a one-parameter group of diffeomorphisms $\varphi_{t}^{c*}$ on $T^{*}M$ satisfying(17)$$\pi_{M}\circ \varphi_{t}^{c*}=\varphi_{t}\circ \pi_{M},$$
where $\pi_{M}$ is the natural projection defined on $T^{*}M$ to M. The vector field $X^{c*}$ on $T^{*}M$, which has the flow $\varphi_{t}^{c*},$ is called the complete cotangent lift of *X* [44]. The infinitesimal version of the Equation (Equation 17) determines $X^{c*}$ as follows,(18)$$T\pi_{M}\circ X^{c*}=X\circ \pi_{M}.$$

The complete cotangent lift $X^{c*}$ is dual to the complete tangent lift $X^{c}$ in the following sense. Taking the duality between elements of the tangent planes and elements of the adjoint cotangent spaces, the duality between the cotangent and tangent lifts must be the same, that is,(19)$$\langle \varphi_{t}^{c*}\left(\Pi \right),v\rangle =\langle \Pi ,\varphi_{-t}^{c}\left(v\right)\rangle .$$

Therefore, we put(20)$$\langle \Pi ,\varphi_{-t}^{c}\left(v\right)\rangle =\Pi_{a}v^{a}-t\Pi_{b}\frac{\partial X^{b}}{\partial x^{a}}v^{a}=\left(\Pi_{a}-t\Pi_{b}\frac{\partial X^{b}}{\partial x^{a}}\right)v^{a}={\bar{\Pi }}_{a}v^{a}=\langle \varphi_{t}^{c*}\left(\Pi \right),v\rangle ,$$
which means that ${\bar{\Pi }}_{a}=\Pi_{a}-t\Pi_{b}\partial X^{b}/\partial x^{a}$. From the perspective of the cotangent bundle $T^{*}M$, a vector field *X* also induces motion in the momentum, which is the complete cotangent lift of vector field to $X(T^{*}Q)$,(21)$$X^{c*}=X^{a}\frac{\partial }{\partial x^{a}}-\frac{\partial X^{b}}{\partial x^{a}}\Pi_{b}\frac{\partial }{\partial \Pi_{a}},$$
see, e.g., the work by the authors of [50]. The complete cotangent lift of *X* expresses how both position on the manifold Q and covectors (one-forms) vary along the motion induced by the field; a diagram summarizing the discussions done so far follows.



We have the following Lemma [30,51].

**Lemma** **1.**
*Maps given in*
(22)$$\begin{array}{c} {}^{c}:X\left(M\right)\to X\left(TM\right):X\to X^{c},{\;}^{c*}:X\left(M\right)\to X\left(T^{*}M\right):X\to X^{c*}. \end{array}$$
*are Lie algebra isomorphism intos, that is,*
(23)$${\left[X,Y\right]}^{c}=\left[X^{c},Y^{c}\right]\;\;\;and\;\;\;{\left[X,Y\right]}^{c*}=\left[X^{c*},Y^{c*}\right],$$
*for all $X,Y\in X\left(M\right).$*


### 2.2. From Jet Bundle to Tangent Bundle

Triple $\left(E,\pi ,M\right)$ denotes a smooth bundle with coordinates $\left(x^{a}\right)$ on the base manifold M and $\left(x^{a},u^{\lambda }\right)$ on the total manifold E. $J^{1}\pi$ is the first order jet manifold associated with $\left(E,\pi ,M\right)$ with induced coordinates $\left(x^{a},u^{\lambda },u_{a}^{\lambda }\right)$. There exist fibrations,$$\begin{array}{c} \pi_{0}:J^{1}\pi \to E:\left(x,u,u_{x}\right)\to \left(x,u\right),\;\pi_{1}:J^{1}\pi \to M:\left(x,u,u_{x}\right)\to \left(x\right) \end{array}$$
of $J^{1}\pi$ on E and M, respectively, [52]. Contact forms(24)$$\vartheta^{a}=du^{\alpha }-u_{a}^{\alpha }dx^{a}$$
determine holonomic sections of the jet bundle fibration.

#### 2.2.1. Holonomic Lifts of Vector Fields

Consider a fiber bundle $\left(E,\pi ,M\right)$. Let *X* be a vector field on M and consider a section σ of the fibration π. The lie derivative (directional derivative) of a smooth function, *F*, defined on the total space, E, with respect to the vector field *X* can be computed by means of σ as follows, $L_{X}\left(F\circ \sigma \right)$. This definition reads the definition of the holonomic lift $X^{hol}$ of the vector field *X* by the identity(25)$$X^{hol}\left(F\right)\circ \sigma :=L_{X}\left(F\circ \sigma \right).$$

Note that in terms of the local coordinates, for $X=X^{a}\partial /\partial x^{a}$, the holonomic lift defined in (Equation 25) is computed to be(26)$$X^{hol}=X^{a}\frac{\partial }{\partial x^{a}}+X^{a}u_{a}^{\lambda }\frac{\partial }{\partial u^{\lambda }}.$$

Note that $X^{hol}$ is not a classical vector field on E, as its coefficients depend on the first order jet bundle. Such kind of sections are called generalized vector fields [52,53,54,55]. In order to justify the term holonomic, we see that the values of $X^{hol}$ at the contact one forms $\vartheta^{a}$ in Equation (Equation 24) vanish identically. For a vector field $Y=Y^{a}\partial /\partial x^{a}+Y^{\alpha }\partial /\partial u^{\alpha }$, we define the holonomic part HY of *Y* as(27)$$HY:={\left(T\pi \circ Y\right)}^{hol}=Y^{a}\frac{\partial }{\partial x^{a}}+Y^{a}u_{a}^{\lambda }\frac{\partial }{\partial u^{\lambda }}.$$

#### 2.2.2. Lie Algebra of Generalized Vector Fields

Assume a fibration $\left(E,\pi ,M\right)$. In general, a generalized vector field on E takes the form of(28)$$\xi =\xi^{a}\left(x\right)\frac{\partial }{\partial x^{a}}+\xi^{\lambda }\left(x,u,u_{x}\right)\frac{\partial }{\partial u^{\lambda }}.$$

The first order prolongation $pr^{1}\xi$ of ξ is defined by(29)$$pr^{1}\xi =\xi +\Phi_{a}^{\lambda }\frac{\partial }{\partial u_{a}^{\lambda }},\;\;\;\Phi_{a}^{\lambda }=D_{x^{a}}\left(\xi^{\lambda }-\xi^{b}u_{b}^{\lambda }\right)+\xi^{b}u_{ba}^{\lambda },$$
where $D_{x^{a}}$ is the total derivative operator with respect to $x^{a}$, and $u_{ba}^{\lambda }$ is an element of the second order jet bundle. The Lie bracket of two first order generalized vector fields ξ and η is the unique first order generalized vector field(30)$${\left[\xi ,\eta \right]}_{pro}=\left(pr^{1}\xi \left(\eta^{a}\right)-pr^{1}\eta \left(\xi^{a}\right)\right)\frac{\partial }{\partial x^{a}}+\left(pr^{1}\xi \left(\eta^{\lambda }\right)-pr^{1}\eta \left(\xi^{\lambda }\right)\right)\frac{\partial }{\partial u^{\lambda }}.$$

If ξ and η are two classical vector fields on E, then ${\left[•,•\right]}_{pro}$ reduces to the Jacobi–Lie bracket of vector fields [54], which results in the following lemma.

**Lemma** **2.**
*The mapping H:Y→HY in (Equation 27), taking a vector field to its holonomic part, is a Lie algebra isomorphism from the space of projectable vector fields into the space of generalized vector fields of order one.*


#### 2.2.3. Vertical Representatives

Note that the holonomic lift of a vector field reads the action of the vector field *X* on the fiber coordinates. So that, for a projectable vector field *Y* on E, to read just the vertical motion, that is, the dynamics governing the sections, one needs to subtract holonomic part inside *Y*, so that the vertical motion is the one obtained by subtracting the holonomic part HY of the vector field from itself [54]. We call this the vertical representative:(31)$$VY=Y-HY=\left(Y^{\alpha }-Y^{a}u_{a}^{\lambda }\right)\frac{\partial }{\partial u^{\lambda }}.$$

Note that VY lies in the kernel of Tπ. On the other hand, the generalized bracket of vertical representatives satisfies(32)$${\left[VY_{1},VY_{2}\right]}_{pro}=V\left[Y_{1},Y_{2}\right]+B\left(Y_{1},Y_{2}\right).$$
B is a vertical-vector valued two-form(33)$$B\left(Y_{1},Y_{2}\right)={\left[HY_{2},VY_{1}\right]}_{pro}-{\left[HY_{1},VY_{2}\right]}_{pro},$$
where the brackets are as in (Equation 30), see [42]. We require that generalized vector fields are projectable [53].

### 2.3. Lie–Poisson Dynamics of Incompressible Systems

Now, we concentrate on the case of the cotangent lift and its vertical representative. For this, we consider the vector bundle $\left(T^{*}M,\pi_{M},M\right)$, and let Π be a section of this bundle. Let *X* be a vector field on *X*, then its holonomic part is(34)$${\left(X^{c*}\right)}^{hol}=X^{hol}=X^{a}\frac{\partial }{\partial x^{a}}+X^{a}\frac{\partial \Pi_{b}}{\partial x^{a}}\frac{\partial }{\partial \Pi_{b}}.$$

The vertical representative of the cotangent lift $X^{c*}$, that is,(35)$$\begin{array}{c} \begin{array}{cc} VX^{c*} & =X^{c*}-{\left(T\pi \circ X^{c*}\right)}^{hol}=X^{c*}-X^{hol}\\ & =-\frac{\partial X^{a}}{\partial x^{b}}\Pi_{a}\frac{\partial }{\partial \Pi_{b}}-X^{a}\frac{\partial \Pi_{b}}{\partial x^{a}}\frac{\partial }{\partial \Pi_{b}}=-{\left(L_{X}\left(\Pi \right)\right)}_{b}\frac{\partial }{\partial \Pi_{b}} \end{array} \end{array}$$
which results with(36)$${\left(VX^{c*}\right)}_{b}=-{\left(L_{X}\left(\Pi \right)\right)}_{b}.$$

We have then the mapping(37)$$V^{c*}:X\left(Q\right)\to VX\left(T^{*}Q\right):X\to VX^{c*},$$
taking a vector field, *X*, on Q to a vertical vector field on $T^{*}Q$, by first taking the cotangent lift, then taking the vertical representative. For the case of complete lifts, the vector-valued two-form B defined in Equation (Equation 33) vanishes. This reads as the following lemma.

**Lemma** **3.**
*The mapping $V^{c*}$ in (Equation 37) is a Lie algebra isomorphism into, that is,*
$$X\left(Q\right)⟷X^{c*}\left(T^{*}Q\right)⟷VX^{c*}\left(T^{*}Q\right).$$


Vertical representative is sought for a vector field $X^{c*}$ on cotangent bundle $T^{*}T^{*}Q$. The cotangent bundle has a natural fiber structure $T^{*}T^{*}Q$ over the base $T^{*}Q$. The vector field thus has a component in the direction $\left(\partial /\partial q^{i},\partial /\partial p^{i}\right)$ as well as in $\left(\partial_{\Pi_{i}},\partial_{\Pi^{i}}\right)$. For a given point $\left(q^{i},p_{i}\right)$, the component of the vector field in the $\left(\partial_{\Pi_{i}},\partial_{\Pi^{i}}\right)$ directions describes evolution of the momentum variables $\left(\Pi_{i},\Pi^{i}\right)$, which can be seen as motion along the fiber attached to $\left(q^{i},p_{i}\right)$. The momentum coordinate can be seen as a function of the point $\left(q^{i},p_{i}\right)$, and the time-evolution equation for $\left(\Pi_{i},\Pi^{i}\right)$ is the vertical component of the vector field. However, as also the point $\left(q^{i},p_{i}\right)$ is subject to motion, evolution of $\left(q^{i},p_{i}\right)$ affects the momentum coordinate $\left(\Pi_{i},\Pi^{i}\right)$. By subtracting this evolution, $X_{h}^{hol}$, from the vertical vector field, $X_{h}^{c*}$, we obtain the vertical representative of the total vector field, which determines the evolution of the momentum variable(38)$${\dot{\Pi }}_{i}=X_{h}^{c*}-X_{h}^{hol}.$$

This is a purely geometric way for extending dynamics on a manifold to the cotangent bundle on the manifold.

#### 2.3.1. Vertical Lifts of One-Forms

Consider the cotangent lift $T^{*}\pi_{M}:T^{*}M\to T^{*}T^{*}M$ of the projection $\pi_{M}:T^{*}M\to M$, and recall the isomorphism $\Omega_{T^{*}M}^{♯}:T^{*}T^{*}M\to TT^{*}M$ associated with the symplectic two-form $\Omega_{T^{*}M}$ on the cotangent bundle $T^{*}M.$ Euler vector field is defined as(39)$$X_{E}:T^{*}M\to TT^{*}M:z\to \Omega_{T^{*}M}^{♯}\circ T^{*}\pi_{M}\left(z\right),$$
which is a vertical-valued vector field, that is, $Im\left(X_{E}\right)\subset ker\left(T\pi_{M}\right)$. The vertical lift(40)$$\alpha^{v}=X_{E}\circ \alpha \circ \pi_{M}:T^{*}M\to TT^{*}M$$
of the one-from α is a vertical-valued vector field on $T^{*}M$ [44]. Taking the coordinates $\left(x^{a},y_{b}\right)$ on $T^{*}M$, the Euler vector field is computed as $X_{E}=-y_{a}\partial /\partial y_{a}$ and the vertical lift of the one-form $\alpha =\alpha_{a}\left(x\right)dx^{a}$ is $\alpha^{v}=-\alpha_{a}\left(x\right)\partial /\partial y_{a}.$

#### 2.3.2. Geometry of Lie–Poisson Equations

Now, we will collect all the geometric structures introduced in this section to arrive at the Lie–Poisson equations in terms of lifts and vertical representatives. Starting with a differential one-form $\Pi =\Pi_{a}\left(x\right)dx^{a}$, an element of the space $\Lambda^{1}\left(M\right)$ of one-form sections on M. Assuming that $\Lambda^{1}\left(M\right)$ is a vector space, its tangent space equals to the product $T\Lambda^{1}\left(M\right)=\Lambda^{1}\left(M\right)\times \Lambda^{1}\left(M\right)$. Let us consider a curve Π(t) in $\Lambda^{1}\left(M\right)$. The time derivative $\dot{\Pi }$ is an element of the tangent space. As we have identified $T_{\Pi }\Lambda^{1}\left(M\right)=\Lambda^{1}\left(M\right)$, the geometric object $\dot{\Pi }$ is a differential one-form $\dot{\Pi }={\dot{\Pi }}_{a}\left(x\right)dx^{a}$ which is in $\Lambda^{1}\left(M\right)$. Now, we compute the vertical lift of this one-form section as explained in (Equation 40), this locally reads that(41)$${\dot{\Pi }}^{v}={\dot{\Pi }}_{a}\left(x\right)\frac{\partial }{\partial \Pi_{a}}.$$

Therefore, a direct observation results in the following proposition.

**Proposition** **1.**
*If the motion of a single particle is governed by a volume preserving vector field X on a manifold M, then the Lie–Poisson equation governing the motion of the continuum consisting of such particles can be written as*
(42)$${\dot{\Pi }}^{v}=VX^{c*}\left(x,\Pi \right).$$


**Proof.** In order to prove this observation, recall the Lie–Poisson equations given in Equation (Equation 9). See that minus of the Lie derivative on the right hand side equals to $VX^{c*}$ that is the vertical representative of the complete cotangent lift of *X*. By employing Equation (Equation 41), one immediately arrives at the required result (Equation 42). □

We remark that in this geometrization, one of the crucial step is to determine the dual space. Here, we are showing this fact explicitly in the following example.

### 2.4. Example: Incompressible Fluid Flow

For an ideal incompressible fluid in a bounded compact region, $Q\subset R^{3}$, the configuration space is the group $Diff_{vol}\left(Q\right)$ of volume preserving diffeomorphisms on Q. The Lie algebra $X_{div}\left(Q\right)$ of $Diff_{vol}\left(Q\right)$ is the algebra of divergence-free vector fields parallel to the boundary of Q, and the dual space $X_{div}^{*}\left(Q\right)$ is the space(43)$$X_{div}^{*}\left(Q\right)=\left\{\left[\Upsilon \right]⊗d^{3}q\in \left(\Lambda^{1}\left(Q\right)/dF\left(Q\right)\right)⊗Den\left(Q\right)\right\},$$
of one-form modulo exact one-form densities on Q. Here, $\left[\Upsilon \right]=\left\{\Upsilon +d\tilde{p}:\tilde{p}\in F\left(Q\right)\right\}\in \Lambda^{1}\left(Q\right)/dF\left(Q\right)$ denotes the equivalence class containing Υ, and the volume three-form $d^{3}q$ is the Euclidean volume on $R^{3}$ [5,56]. Let $\left(x^{a},\Upsilon_{b}\right)$ be induced coordinates and $X=X^{a}\partial /\partial x^{a}$ be a divergence-free vector field. Then,(44)$$VX^{c*}=\left(-\Upsilon_{b}\frac{\partial X^{b}}{\partial x^{a}}-X^{a}\frac{\partial \Upsilon_{b}}{\partial x^{a}}\right)\frac{\partial }{\partial \Upsilon_{a}},$$
according to Equation (Equation 36), and the equations of motion for the dynamics generated by $VX^{c*}$ are(45)$$\frac{\partial \left[\Upsilon \right]}{\partial t}=-L_{X}\left[\Upsilon \right].$$

For a generic element $\Upsilon +d\tilde{p}\in \left[\Upsilon \right],$ Equation (Equation 45) becomes Euler’s equations for ideal fluid, that is $\partial \Upsilon /\partial t+L_{X}\Upsilon =dp.$ If the dual space $X_{div}^{*}\left(Q\right)$ is identified with exact two-forms by $\left[\Upsilon \right]\to d\Upsilon =\omega \in \Lambda^{2}\left(Q\right)$, then Equation (Equation 45) becomes the Euler’s equation in vorticity form $\partial \omega /\partial t+L_{X}\omega =0.$

### 2.5. Example: Vlasov’s Plasma

Configuration space of collisionless and non-relativistic plasma motion is group $Diff_{can}\left(T^{*}Q\right)$ of canonical diffeomorphisms on the phase space $T^{*}Q$ of configuration manifold $Q\subset R^{3}$ of individual charged particles [2,57,58].

#### 2.5.1. Lie Algebra of the Canonical Diffeomorphisms

Lie algebra of the group is space $X_{ham}\left(T^{*}Q\right)$ of Hamiltonian vector fields on $T^{*}Q$ equipped with minus of the Jacobi–Lie bracket [•,•]. We can identify the space $X_{ham}\left(T^{*}Q\right)$ with the space of nonconstant smooth functions (more terminologically nonconstant Hamiltonian functions) on $T^{*}Q$ equipped with the canonical Poisson bracket {•,•} as the Lie algebra bracket, that is,(46)$$\left(F\left(T^{*}Q/R\right),\left\{•,•\right\}\right)\to \left(X_{ham}\left(T^{*}Q\right),-\left[•,•\right]\right):h\to X_{h}.$$

This is a manifestation of the identity(47)$$-\left[X_{h},X_{f}\right]=X_{\left\{h,f\right\}}.$$

#### 2.5.2. The Dual Space

The identification (Equation 47) reads that Lie algebra of the canonical diffeomorphism can also be considered as $F(T^{*}Q)/R$. In this case, the dual space of the Lie algebra is the space $Den(T^{*}Q)$ of densities on $T^{*}Q$. By fixing the symplectic volume $\Omega_{Q}^{3}$, we can further identify the dual space with the smooth functions $F(T^{*}Q)$. This fits the classical approach, where elements in the function space are accommodated as the plasma density functions. We are now returning to the very first definition of the Lie algebra consisting of the Hamiltonian vector fields, and try to define a dual to that space consisting of the differential one-forms.

**Lemma** **4.**
*The following identity holds,*
(48)$$\begin{array}{c} \int_{T^{*}Q}\langle X_{h}\left(z\right),\Pi \left(z\right)\rangle d\mu \left(z\right)=\int_{T^{*}Q}h\left(div_{\Omega_{T^{*}Q}}\Pi^{♯}\right)d\mu , \end{array}$$
*where $\Omega_{T^{*}Q}^{♯}:\Pi \to \Pi^{♯}$ is induced from the symplectic two-form $\Omega_{T^{*}Q}$.*


**Proof.** With this definition of the dual space the $L_{2}$-pairing of the Lie algebra and its dual becomes nondegenerate provided we take the symplectic volume $d\mu =\Omega_{T^{*}Q}^{3}$ in(49)$$\begin{array}{ccc} \int_{T^{*}Q}\langle X_{h}\left(z\right),\Pi \left(z\right)\rangle d\mu \left(z\right) & = & -\int_{T^{*}Q}\langle dh,\Pi^{♯}\rangle d\mu =-\int_{T^{*}Q}i_{\Pi^{♯}}\left(dh\right)d\mu \\ & = & -\int_{T^{*}Q}dh\wedge i_{\Pi^{♯}}d\mu =\int_{T^{*}Q}hd\left(i_{\Pi_{f}^{♯}}d\mu \right)\\ & = & \int_{T^{*}Q}h\left(div_{\Omega_{T^{*}Q}}\Pi^{♯}\right)d\mu , \end{array}$$
where we have applied integration by parts in the second line ([30], internet supplement). The calculation can be also carried out in the Darboux coordinates using $X_{h}=L\cdot dh$, where *L* is the Poisson bivector (inverse of $\Omega_{Q}$). □

**Proposition** **2.**
*The $L^{2}$-dual space of the Lie algebra $X_{ham}\left(T^{*}Q\right)$ of Hamiltonian vector fields is*
(50)$$X_{ham}^{*}\left(T^{*}Q\right)=\left\{\Pi ⊗d\mu \in \Lambda^{1}\left(T^{*}Q\right)⊗Den\left(T^{*}Q\right):div_{\Omega_{Q}}\Pi^{♯}\neq 0\right\}.$$


**Proof.** Then, the dual of the Lie algebra isomorphism $h\to X_{h}$ is(51)$$\Pi \to div_{\Omega_{Q}}\Pi^{♯}$$
and it is a momentum map. Notice that the operator $div_{\Omega_{Q}}^{♯}$ takes the one-form of a real valued function. In Darboux’s coordinates $\left(q^{i},p_{i}\right)$ on $T^{*}Q$, if $\Omega_{T^{*}Q}=dq^{i}\wedge dp_{i}$ and $\Pi_{f}=\Pi_{i}dq^{i}+\Pi^{i}dp_{i}\;$, then(52)$$f\left(z\right)=div_{\Omega_{Q}}\Pi^{♯}\left(z\right)=\frac{\partial \Pi^{i}\left(z\right)}{\partial q^{i}}-\frac{\partial \Pi_{i}\left(z\right)}{\partial p_{i}},$$
defines the plasma density function. This calculation leads us to add a subscript *f* to the notation of the one-form section $\Pi_{f}$, so that we have $div_{\Omega_{Q}}\Pi_{f}^{♯}=f$. □

#### 2.5.3. Momentum-Vlasov Equations

We start with the total energy function $h=p^{2}/2m+e\phi_{\Pi }$ of a single particle. Here, $\phi_{\Pi }$ is the potential. Locally, the Hamiltonian vector field for the Hamiltonian function *h* is computed to be(53)$$X_{h}=\delta^{ij}\frac{p_{j}}{m}\frac{\partial }{\partial q^{i}}-e\frac{\partial \phi_{\Pi }}{\partial q^{i}}\frac{\partial }{\partial p_{i}}\in X\left(T^{*}Q\right).$$

In Darboux’ coordinates $\left(q^{i},p_{i},\Pi_{i},\Pi^{i}\right)$ on $T^{*}T^{*}Q$, the complete cotangent lift of $X_{h}$ reads(54)$$X_{h}^{c*}=X_{h}-\delta^{ij}\frac{1}{m}\Pi_{i}\frac{\partial }{\partial \Pi^{j}}+e\Pi^{j}\frac{\partial^{2}\phi_{\Pi }}{\partial q^{j}\partial q^{i}}\frac{\partial }{\partial \Pi_{i}}.$$

The vertical representative of the cotangent lift $X_{h}^{c*}$,(55)$$VX_{h}^{c*}=\left(e\Pi^{j}\frac{\partial^{2}\phi }{\partial q^{j}\partial q^{i}}-X_{h}\left(\Pi_{i}\right)\right)\frac{\partial }{\partial \Pi_{i}}-\left(\frac{1}{m}\Pi_{j}\delta^{ji}+X_{h}\left(\Pi^{i}\right)\right)\frac{\partial }{\partial \Pi^{i}},$$
is a vertical valued generalized vector field of order 1. Vertical lift of the one-from $\Pi_{f}=\Pi_{i}dq^{i}+\Pi^{i}dp_{i}$ is a vertical vector field $\Pi_{f}^{v}=\Pi^{i}\partial_{q^{i}}-\Pi_{i}\partial p_{i}$ on $T^{*}T^{*}Q$, [44]. Hence, m-Vlasov Equations (Equation 56) can be recast in the form$${\dot{\Pi }}_{f}^{v}=VX_{h}^{c*},$$
that is given explicitly by(56)$$\begin{array}{cc} {\dot{\Pi }}_{i} & =e\Pi^{j}\frac{\partial^{2}\phi }{\partial q^{j}\partial q^{i}}-X_{h}\left(\Pi_{i}\right)\\ {\dot{\Pi }}^{i} & =-\frac{1}{m}\Pi_{j}\delta^{ji}-X_{h}\left(\Pi^{i}\right), \end{array}$$

**Vlasov Equation.** Let us recall the identity (Equation 42) and apply it in the present case. The momentum-Vlasov equations can be compactly written as(57)$${\dot{\Pi }}_{f}=VX_{h}^{c*}\left(\Pi_{f}\right)=-L_{X_{h}}\Pi_{f}.$$

**Lemma** **5.**
*The operator $div_{\Omega_{Q}}^{♯}$ and the Lie derivative $L_{X_{h}}$ commute for Hamiltonian vector fields $X_{h}$, that is*
(58)$$div_{\Omega_{Q}}^{♯}\left(L_{X_{h}}\Pi_{f}\right)=L_{X_{h}}\left(div_{\Omega_{Q}}^{♯}\left(\Pi_{f}\right)\right).$$


**Proof.** Recall the calculation in (Equation 49) and apply the present case as follows(59)$$\int_{T^{*}Q}\langle \left(L_{X_{h}}\Pi_{f}\right),X_{k}\rangle d\mu \left(z\right)=\int_{T^{*}Q}div_{\Omega_{Q}}^{♯}\left(L_{X_{h}}\Pi_{f}\right)kd\mu \left(z\right).$$On the other hand, we have an integration that by part reads,(60)$$\begin{array}{cc} \int_{T^{*}Q}\langle \left(L_{X_{h}}\Pi_{f}\right),X_{k}\rangle d\mu \left(z\right)\; & =-\int_{T^{*}Q}\langle \Pi_{f},L_{X_{h}}X_{k}\rangle d\mu \left(z\right)=\int_{T^{*}Q}\langle \Pi_{f},X_{\left\{h,k\right\}}\rangle d\mu \left(z\right)\\ & =\int_{T^{*}Q}div_{\Omega_{Q}}^{♯}\left(\Pi_{f}\right)\left\{h,k\right\}d\mu \left(z\right)=\int_{T^{*}Q}\left\{div_{\Omega_{Q}}^{♯}\left(\Pi_{f}\right),h\right\}kd\mu \left(z\right)\\ \; & =\int_{T^{*}Q}L_{X_{h}}\left(div_{\Omega_{Q}}^{♯}\left(\Pi_{f}\right)\right)k\;d\mu \left(z\right). \end{array}$$Comparing the first and the second calculations in the proof for an arbitrary function *k* the proof is completed. □

When the dual mapping in Equation (Equation 51) is employed, the momentum-Vlasov equations then turn to(61)$$div_{\Omega_{Q}}^{♯}\left({\dot{\Pi }}_{f}\right)=-div_{\Omega_{Q}}^{♯}\left(L_{X_{h}}\Pi_{f}\right).$$

If, in particular, the dualization is determined by the $L^{2}$ pairing on the function space, we arrive at(62)$$div_{\Omega_{Q}}^{♯}\left(L_{X_{h}}\Pi_{f}\right)=L_{X_{h}}\left(div_{\Omega_{Q}}^{♯}\left(\Pi_{f}\right)\right)=L_{X_{h}}\left(f\right)=\left\{f,h\right\}.$$

This reads the Eulerian dynamics in density variables, that is, Vlasov’s equation,(63)$$\dot{f}=div_{\Omega_{Q}}^{♯}\left({\dot{\Pi }}_{f}\right)=-div_{\Omega_{Q}}^{♯}\left(L_{X_{h}}\Pi_{f}\right)=-\left\{f,h\right\}.$$

If *h* is the total energy of a single particle, we have(64)$$\frac{\partial f}{\partial t}+\frac{1}{m}p_{i}\frac{\partial f}{\partial q^{i}}-e\frac{\partial^{2}\phi }{\partial q^{i}}\frac{\partial f}{\partial p_{i}}=0.$$

In the work by the authors of [57], the accompanying Poisson equation(65)$$\nabla_{q}^{2}\phi_{f}\left(q\right)=-e\int f\left(q,p\right)d^{3}p$$
has been obtained by a momentum mapping coming from the gauge symmetry of the Hamiltonian dynamics.

### 2.6. Example: Averaged 2D-Euler Equation

In the previous example we have employed $L^{2}$ pairing of the functions, that is simply multiply-and-integrate. This determines the structure of the Lie–Poisson equation. Let us consider that *Q* equals to R with coordinates *x*, so that the cotangent bundle turns out to be $T^{*}Q=R^{2}$ with coordinates (x,y). Now, we change the pairing to the Sobolev $H^{1}$-pairing, given by(66)$${\langle f_{1},f_{2}\rangle }_{H^{1}}=\int f_{1}f_{2}\;dxdy+\lambda^{2}\int \nabla f_{1}\cdot \nabla f_{2}\;dxdy,$$
where λ is a real parameter. Here, ∇f is the gradient of *f*. After applying integration by-parts to the second term and omitting the total divergence terms we write $H^{1}$-pairing in terms of the $L^{2}$-pairing as follows(67)$${\langle f_{1},f_{2}\rangle }_{H^{1}}=\langle \left(1-\lambda^{2}\Delta \right)f_{1},f_{2}\rangle =\int f_{2}\left(1-\lambda^{2}\Delta \right)f_{1}\;dxdy.$$

In this framework, the momentum map defined in (Equation 51) takes the form of(68)$$\Pi \left(z\right)\to div_{\Omega_{Q}}\Pi^{♯}\left(z\right)=\left(1-\lambda^{2}\Delta \right)f.$$

To see that, consider the following equalities(69)$$\int_{T^{*}Q}\langle X_{h}\left(z\right),\Pi \left(z\right)\rangle \;dxdy\left(z\right)=\int_{T^{*}Q}h\left(div_{\Omega_{T^{*}Q}}\Pi^{♯}\right)\;dxdy=\int h\left(1-\lambda^{2}\Delta \right)f\;dxdy,$$
where we have employed Lemma (4) for the case of $T^{*}Q=R^{2}$ and used the Sobelev metric (Equation 67) on the function space. Note that in this case we omit the subscript in order not to mix this mapping with the one in (Equation 51). Let us apply the momentum mapping (Equation 68) to both sides of the equation $\dot{\Pi }=VX_{h}^{c*}$. From the left hand side, one arrives at(70)$$div_{\Omega_{Q}}^{♯}\left(\dot{\Pi }\right)=\left(1-\lambda^{2}\Delta \right)\dot{f}$$
for the left hand side, one computes(71)$$div_{\Omega_{Q}}^{♯}\left(VX_{h}^{c*}\right)=-div_{\Omega_{Q}}^{♯}\left(L_{X_{h}}\Pi \right)=-L_{X_{h}}div_{\Omega_{Q}}^{♯}\Pi =-L_{X_{h}}\left(\left(1-\lambda^{2}\Delta \right)f\right)$$
where we have used (Equation 36) in the first equality and used the commutation relation (5) in the second equality. Therefore, we have(72)$$\left(1-\lambda^{2}\Delta \right)\dot{f}=-\left\{\left(1-\lambda^{2}\Delta \right)f,h\right\}.$$

Assume, in particular, that f=Ω is the vorticity of an ideal inviscid incompressible homogeneous fluid and h=Ψ is the stream function determined by the equation Ω=ΔΨ; then, this system turns out to be an averaged 2D-Euler equation, [59,60](73)$$\left(1-\lambda^{2}\Delta \right)\dot{\Omega }+\left\{\left(1-\lambda^{2}\Delta \right)\Omega ,\Psi \right\}=0.$$

## 3. Generalized Complete Cotangent Lift

Before starting to elaborate the fluid theories and the kinetic moments of the plasma dynamics, we study some geometrical arguments motivating from [47].

### 3.1. Schouten Concomitant

Direct product $TQ={⊕}_{n=0}^{\infty }T^{n}Q$ of spaces $T^{n}Q$ of symmetric contravariant tensor fields on a tmanifold $Q\subset R^{3}$ of all orders constitutes an infinite dimensional vector space. In a local coordinate system $\left\{q^{i}\right\}$ on Q, an element of TQ can be written in the form of(74)$$X={⊕}_{n=0}^{\infty }X^{n}={⊕}_{n=0}^{\infty }X^{i_{1}i_{2}\ldots i_{n}}\left(q\right)\partial_{q^{i_{1}}}⊗\ldots ⊗\partial_{q^{i_{n}}},$$
where $X^{n}\in T^{n}Q$ is a symmetric contravariant tensor field of order *n* and $X^{i_{1}i_{2}\ldots i_{n}}$ are real valued coefficient functions. $T^{0}Q$ is the space $F\left(Q\right)$ of smooth functions and $T^{1}Q$ is the space $X\left(Q\right)$ of vector fields on Q. Schouten concomitant(75)$${\left[X,Y\right]}_{SC}={⊕}_{n=0}^{\infty }{⊕}_{m=0}^{\infty }{\left[X^{n},Y^{m}\right]}_{SC}={⊕}_{n=0}^{\infty }{⊕}_{m=0}^{\infty }Z^{n+m-1}$$
is a Lie algebra structure on the space TQ [39,49,61]. Here $X^{n},$$Y^{m}$ and $Z^{n+m-1}$ are contravariant tensor fields of orders *n*, *m* and n+m−1, respectively. The coefficient functions of $Z^{n+m-1}$ in terms of those $X^{n}$ and $Y^{m}$ are$$Z^{i_{1}\ldots i_{n+m-1}}=nX^{i_{m+1}\ldots i_{m+n-1}l}\frac{\partial Y^{i_{1}\ldots i_{m}}}{\partial q^{l}}-mY^{i_{n+1}\ldots i_{n+m-1}l}\frac{\partial X^{i_{1}i_{2}\ldots i_{n}}}{\partial q^{l}}.$$

#### 3.1.1. Lie Subalgebras of Schouten Algebra

For the zeroth-order tensors, that is, for the space of smooth functions $F\left(Q\right)$, Schouten concomitant reduces to the trivial Poisson bracket of functions on Q. So that $F\left(Q\right)$ is a subalgebra of T(Q). For the first order tensors, that is, for the space of smooth vectors $X\left(Q\right)$, the concomitant turns out to be the Jacobi–Lie bracket of vector fields. For instance, when $X_{1}$ and $X_{2}$ are vector fields in the classical sense, then the coefficient function becomes(76)$$Z^{i}=X_{1}^{i}\frac{\partial X_{2}^{j}}{\partial q^{i}}-X_{2}^{i}\frac{\partial X_{1}^{j}}{\partial q^{i}}={\left[X_{1},X_{2}\right]}^{i},$$
which is the Jacobi–Lie bracket (commutator) of the two vector fields. So, $X\left(Q\right)$ is another subalgebra of T(Q). A semidirect product(77)$$s:=\sum_{k=0}^{1}\;TQ=F\left(Q\right)⋊X\left(Q\right)$$
of the subalgebras is another subalgebra of T(Q). In this case, the Schouten concomitant becomes(78)$${\left[\left(\rho_{1},X_{1}\right),\left(\rho_{2},X_{2}\right)\right]}_{SC}=\left(L_{X_{1}}\rho_{2}-L_{X_{2}}\rho_{1},\left[X_{1},X_{2}\right]\right),$$
where $L_{X}\rho$ is the directional derivative of the function ρ in the direction of *X* and $\left[X,Y\right]$ is the Jacobi–Lie bracket of vector fields *X* and *Y*.

Note also that the complement of the vector space s in TQ is closed under the Schouten concomitant as well. We denote this subalgebra by(79)$$n:=\sum_{k=2}^{\infty }\;T^{k}Q.$$

To see this, herein, we only recall the graded character of the Schouten bracket. The least order tensors in n are of order 2, and the Schouten bracket of two such tensors is a third-order tensor, so it is in n. We record here the decomposition TQ≅s⊕n. That is, we can write any generalized tensor field X in the Lie algebra TQ in the form of(80)X=(ρ,X)⊕X,
where the first factor (ρ,X) lives in the subalgebra s whereas the second factor $X=\sum_{n=2}^{\infty }X^{n}$ is in the Lie subalgebra n.

**Remark** **1.**
*This splitting can be interpreted in the context of Grad hierarchy of kinetic theory. The s subalgebra represents fluid mechanics (compressible and isentropic), whereas subalgebra n represents the higher moments of the Grad hierarchy. The splitting then means that closed evolution equations can be formulated within s, within n or within the whole TQ. The first case corresponds to fluid mechanics (Euler equations), the second to dynamics of higher moments only (the reducing dynamics approaching fluid mechanics), and the third case is equivalent to solving the whole Vlasov (or Boltzmann) equation, see the work by the authors of [62] for details.*


#### 3.1.2. Matched Pair (Bicross Product) Realization of TQ

Let us examine algebraic foundation of the decomposition of TQ≅s⊕n presented in the previous paragraph. To this end, we find the possible actions of s and n onto each other by computing the Schouten bracket$$\begin{array}{cc} {\left[X^{k},\left(\rho ,X\right)\right]}_{SC} & ={\left[X^{k},\rho \right]}_{SC}+{\left[X^{k},X\right]}_{SC}={\left[X^{k},\rho \right]}_{SC}-L_{X}X^{k} \end{array}$$
where $X^{k}$ is a symmetric tensor field of order *k* greater then or equal to 2. Here, $L_{X}X^{k}$ is the Lie derivative of $X^{k}$ in the direction of *X*, whereas(81)$${\left[X^{k},\rho \right]}_{SC}=kX^{i_{1}\ldots i_{k-1}\ell }\rho_{,\ell }\partial q^{i_{1}}\ldots \partial q^{i_{k-1}}.$$

Therefore, if *k* is strictly greater than 2, then the order of the tensor ${\left[X^{k},\rho \right]}_{SC}$ is strictly greater than 1 and it is an element of n. Otherwise, that is for k=2, ${\left[X^{k},\rho \right]}_{SC}$ is a first order tensor field (that is a vector field in the classical sense), so that ${\left[X^{2},\rho \right]}_{SC}$ is in s. If the algebraic structure of the decomposition TQ≅s⊕n were a direct product, then two components (ρ,X) and $X=\sum_{k=2}^{\infty }X^{k}$, c.f. Equation (Equation 80), would be orthogonal with respect to Schouten bracket. However, we have proved that this is not the case for TQ. If the algebraic structure of the decomposition TQ≅s⊕n were a semidirect product, then the Schouten bracket of an element of n and an element of s would lie in one of the subspaces. But we have shown, this is not the case for TQ either. Instead, the bracket ${\left[X,\left(\rho ,X\right)\right]}_{SC}$ results in some terms lying in s and some other terms lying in n. In light of the work by the authors of [63], this observation manifests that the total space TQ can be realized a matched pair of its subalgebras s and n. A matched pair is a generalization of the semidirect product in the sense that there exists mutual nontrivial actions of s and n on to the each other. If there is only a one-sided action of s on n or vice versa, the matched pair becomes a semidirect product. The mutual actions are computed from the identity(82)$${\left[X,\left(\rho ,X\right)\right]}_{SC}=X⊳\left(\rho ,X\right)⊕X⊲\left(\rho ,X\right).$$

Here, the first term X⊳(σ,X) is the left action of X∈n on (σ,X)∈s, and the second term X⊲(σ,X) is the right action of (σ,X) on X. Accordingly, we compute the mutual actions as follows,(83)$$\begin{array}{cc} \; & ⊳:s⊗n\mapsto s,\;X⊳\left(\rho ,X\right)=(0,{\left[X^{2},\rho \right]}_{SC}), \end{array}$$
(84)$$\begin{array}{cc} \; & ⊲:s⊗n\mapsto n,\;X⊲\left(\rho ,X\right)=\sum_{n=2}^{\infty }\left({\left[X^{n+1},\rho \right]}_{SC}-L_{X}X^{n}\right). \end{array}$$

We summarize this matched pair decomposition of TQ in the following proposition which says that TQ is a matched pair Lie algebra.

**Proposition** **3.**
*The space TQ of symmetric contravariant tensor fields can be written as a matched pair product of its Lie subalgebras s and n exhibited in (Equation 77) and (Equation 79), that is,*
(85)TQ≅s⋈n,
*where the mutual actions are in Equations (Equation 83) and (Equation 84).*


**Remark** **2.**
*Kolmogorov cascade. Simple fluids are fluids with an internal structure that remains unchanged during the time evolution. Experience shows that for such fluids the level of classical fluid mechanics (i.e., the level on which the hydrodynamic fields play the role of state variables) is autonomous. This experimental observation is compatible with the result (Equation (Equation 85)). The question now is whether there are autonomous mesoscopic levels with a larger, but finite, number of Grad fields. The result (Equation (Equation 85)) indicates that the answer is negative. There are two additional observations supporting the negative answer and thus indirectly also (Equation (Equation 85)).*

*First, it is the dissipation added by Boltzmann to the Hamiltonian kinetic equation. The Boltzmann dissipation, which indeed drives solutions to the level of fluid mechanics, is supported by an independent and a very strong physical argument, namely, that the principal culprit of the disorder created in the gas particle trajectories are the binary collisions. There does not seem to be any other physical process in the gas that would drive solutions to a higher order mesoscopic level.*

*The second is the observation of turbulent flows. When the external force driving the laminar flow increases the macroscopic order of the laminar flow starts to break up. Vortices start to emerge. The vortices can be regarded as an internal structure characterized by higher order Grad moments (i.e., by the fields that have the physical interpretation of higher order velocity correlations). Fluids subjected to a turbulent flow can be thus regarded as complex fluids (i.e., fluids in which the time evolution of the internal structure is coupled to the time evolution of the hydrodynamic fields). Observations of turbulent flows show that the break up continues into smaller and smaller vortices until they completely disappear and become a part of the molecular motion. This observation is known as Kolmogorov cascade. If there was an autonomous mesoscopic level with n higher order Grad moments then, when the vortices would reach the size corresponding to the n-moment, the turbulent flow would become a flow that, in the context of the n -Grad moment level, would appear laminar. We would then expect that the continuation of the break up would appear as an onset of an n-order turbulence, i.e., a turbulence emerging on the n-Grad moment level. In other words, the Kolmogorov cascade would have a more complex dependence on the driving force than the one observed.*


Due to the mutual interactions existing in matched pair products, they can be considered as a generalization of the semidirect products. The Lagrangian and Hamiltonian dynamics on these systems are available [64,65], and the discrete matched pair dynamics of Lie groupoids are discussed in the work by the authors of [66].

#### 3.1.3. Lie Group Underlying s

Let us denote the right and left actions of group $Diff\left(Q\right)$ of diffeomorphisms on Q by(86)$$\psi^{R,L}:Diff\left(Q\right)\times Q\to Q:\left(g,q\right)\to \psi_{g}^{R,L}\left(q\right),$$
respectively. Infinitesimal generators of the diffeomorphisms $\psi_{g}^{R,L}$ are vector fields $X^{R,L}$ on Q, respectively. By fixing a point q∈Q in (Equation 86), we obtain induced mappings $\psi_{q}^{R,L}$ from the group $Diff\left(Q\right)$ to the manifold Q. Actions of $Diff\left(Q\right)$ on space $F\left(Q\right)$ of smooth functions are given by means of pull back operation, that is,(87)$$\sigma^{R,L}\left(g,\phi \right)={\left(\psi_{g^{-1}}^{R,L}\right)}^{*}\phi \;\forall \phi \in F\left(Q\right).$$

Infinitesimal generators $X_{F\left(Q\right)}^{R,L}$ of the transformations $\sigma_{g}^{R,L}$ are vector fields on $F\left(Q\right)$, that is linear transformations on $F\left(Q\right)$. Explicitly, we compute the generators as follows,(88)$$\begin{array}{ccc} X_{F\left(Q\right)}^{R,L}\left(\phi \right)\left(q\right) & = & {\left.\frac{d}{dt}\right|}_{t=0}\sigma^{R,L}\left(g\left(t\right),\phi \right)\left(q\right)={\left.\frac{d}{dt}\right|}_{t=0}\phi \circ \psi^{R,L}\left(g^{-1}\left(t\right),q\right)\\ & = & T_{q}\phi \circ {\left.\frac{d}{dt}\right|}_{t=0}\psi^{R,L}\left(g^{-1}\left(t\right),q\right)=-T_{q}\phi \circ X^{R,L}\left(q\right)\\ & = & -L_{X^{R,L}}\phi \left(q\right) \end{array}$$
where Tϕ is the tangent mapping of ϕ. $L_{X^{R,L}}$ denote the Lie derivative operator and in this case, they are directional derivatives of ϕ in the directions of $X^{R,L}\left(q\right)$, respectively. Actions $\sigma^{R,L}$ of $Diff\left(Q\right)$ on $F\left(Q\right)$ in Equation (Equation 87) define respective semidirect product structures(89)$$\begin{array}{ccc} \left(g_{1},\phi_{1}\right)⋊\left(g_{2},\phi_{2}\right) & = & \left(g_{1}g_{2},\phi_{1}+\sigma^{L}\left(g_{1},\phi_{2}\right)\right) \end{array}$$
(90)$$\begin{array}{ccc} \left(g_{1},\phi_{1}\right)⋉\left(g_{2},\phi_{2}\right) & = & \left(g_{1}g_{2},\phi_{1}+\sigma^{R}\left(g_{1}^{-1},\phi_{2}\right)\right) \end{array}$$
on the product manifolds $S^{R,L}=Diff\left(Q\right){⋉}^{R,L}F\left(Q\right)$. Here, the superscripts of *S* denote which of the actions $\sigma^{R,L}$ has been chosen. Identity elements for both of the group structures $S^{R,L}$ are the same and it is $\left(e,0\right),$ where *e* is the identity in $Diff\left(Q\right)$. The elements of tangent space$$T_{\left(g,\rho \right)}S^{R,L}=T_{g}Diff\left(Q\right)\times F\left(Q\right)$$
at $\left(g,\phi \right)$ are given by two-tuples $\left(X_{g}^{R,L},\nu \right)$. Here, $X_{g}^{R,L}$ are material velocity fields satisfying the identities $\tau_{Q}\circ X_{g}^{R,L}=\psi_{g}^{R,L},$ where $\tau_{Q}$ is the tangent bundle projection and $\psi_{g}^{R,L}$ are diffeomorphisms in Equation (Equation 86). The second term ν is a function on Q, that is an element of $F\left(Q\right)$. Tangent space $T_{\left(e,0\right)}S^{R,L}$ at the identity $\left(e,0\right)$ is the product space $X\left(Q\right)\times F\left(Q\right)$. It is the underlying vector space for both of the Lie algebras $s^{R,L}$ induced from the group structures $S^{R,L}$, respectively. The Lie algebra bracket on $s^{R}$ is the subalgebra structure given in (Equation 78) whereas the bracket for $s^{L}$ needs a minus sign.

### 3.2. Lifts of Tensor Fields to the Cotangent Bundle

#### 3.2.1. Tensors to Functions

Let $X^{k}$ be a contravariant tensor field of order *k*. Due to the canonical inclusion $T^{k}Q↪T^{k}\left(T^{*}Q\right)$, we may assume $X^{k}$ as a tensor field on the cotangent bundle $T^{*}Q$. Using the canonical one-form $\theta =p_{i}dq^{i}$, we define a mapping from $T^{k}Q$ to the space $F\left(T^{*}Q\right)$ of smooth functions on $T^{*}Q$ by contracting the contravariant tensor $X_{n}\in T^{n}Q$ with *n*-th tensor power $\theta_{T^{*}Q}^{n}=\theta_{T^{*}Q}⊗\ldots ⊗\theta_{T^{*}Q}$ of the canonical one form $\theta_{T^{*}Q}$, that is,(91)$$\begin{array}{ccc} X^{k}\to h_{X^{k}} & = & \theta_{T^{*}Q}^{n}\left(X^{n}\right)=\langle p_{i_{1}}\ldots p_{i_{k}}dq^{i_{1}}\ldots dq^{i_{k}},X^{j_{1},\ldots ,j_{k}}\frac{\partial }{\partial q^{j_{1}}}\ldots \frac{\partial }{\partial q^{j_{k}}}\rangle \\ & = & p_{i_{1}}\ldots p_{i_{k}}X^{i_{1},\ldots ,i_{k}} \end{array}$$

We extend the operation given in Equation (Equation 91) to the product space TQ. For $X=⊕X^{n}\in TQ$ we define a function $h_{X}$ on $T^{*}Q$ as the sum(92)$$TQ\to F\left(T^{*}Q\right):X\to h_{X}=\sum_{k=0}^{\infty }h_{X_{k}},$$

Ref. [67]. This infinite sum may be considered as the Taylor expansion of the function $h_{X}$ in terms of the p−polynomials. A straightforward calculation proves the following lemma. For this result, we refer to the work by the authors of [47].

**Lemma** **6.**
*The map $X\to h_{X}$ is a Lie algebra anti-homomorphism, that is,*
$$h_{{\left[X,Y\right]}_{SC}}=-\left\{h_{X},h_{Y}\right\},$$
*where the bracket at the left hand side is the Schouten concomitant of contravariant tensors whereas the bracket at the right hand side is the canonical Poisson bracket of functions on $T^{*}Q$.*


#### 3.2.2. Generalized Complete Cotangent Lift (GCCL)

We further define the following operation, which we call the generalized complete cotangent lift (and denoted as GCCL) from $X^{k}$, to the space of Hamiltonian vector fields(93)$$\mathrm{GCCL}:T^{k}Q\to X_{ham}\left(T^{*}Q\right):X^{n}\to X_{h_{X^{n}}},$$
which take a contravariant tensor field $X^{n}$ on Q to Hamiltonian vector field, corresponding to the Hamiltonian function defined by Equation (Equation 91), c.f., the work by the authors of [68]. In Darboux’ coordinates $\left(q^{i},p_{i}\right)$, the Hamiltonian function $h_{X^{k}}$ is a p−polynomial, and GCCL of $X^{k}$ is thus(94)$$\mathrm{GCCL}\left(X^{n}\right)=kp_{i_{1}}p_{i_{2}}\ldots p_{i_{k-1}}X^{i_{1}\ldots i_{k-1}l}\partial_{q^{l}}-p_{i_{1}}p_{i_{2}}\ldots p_{i_{k}}\frac{\partial X^{i_{1}i_{2}\ldots i_{k}}}{\partial q^{l}}\partial_{p_{l}}.$$

Notice that GCCL of a vector field $X^{1}=X$ is exactly the same as the complete cotangent lift of (Equation 21). GCCL is indeed a generalization of the complete cotangent lift. More generally, the generalized complete cotangent lift of $X=⊕X^{n}$ is defined as(95)$$\mathrm{GCCL}:TQ\to X_{ham}\left(T^{*}Q\right):X={⊕}_{n=0}^{\infty }X^{n}\to \sum_{n=0}^{\infty }\mathrm{GCCL}\left(X^{n}\right).$$

Using the identity in (Equation 47) and Lemma (6), we arrive at the following equalities,$$\left[X^{c*},Y^{c*}\right]=\left[X_{h_{X}},X_{h_{Y}}\right]=-X_{\left\{h_{X},h_{Y}\right\}}=X_{h_{{\left[X,Y\right]}_{SC}}}={\left[X,Y\right]}_{SC}^{c*},$$
which enable us to state the following proposition.

**Proposition** **4.***The generalized complete cotangent lift* GCCL *operation in (Equation 95) is an injective Lie algebra homomorphism, that is*(96)$$\mathrm{GCCL}{\left[X,Y\right]}_{SC}=\left[\mathrm{GCCL}(X),\mathrm{GCCL}(Y)\right],$$
*where ${\left[•,•\right]}_{SC}$ is the Schouten concomitant of tensor fields in Equation (Equation 75) and $\left[•,•\right]$ is the Jacobi–Lie bracket of vector fields on $T^{*}Q$.*

#### 3.2.3.  GCCL on the Subalgebras of T(Q)

We have presented four subalgebras of the the Lie algebra T(Q) equipped with Schouten bracket. They are the space of smooth functions F(Q), the space X(Q) of vector fields, their semidirect product s in (Equation 77), and the complement of the space s denoted by n in (Equation 79). On F(Q), the mapping in Equation (Equation 91) reduces to the natural inclusion of $F\left(Q\right)\;$ into the space $F\left(T^{*}Q\right)$, $h_{\rho }=\rho$, see also [30]. GCCL then takes the particular form(97)$$\rho \to X_{\rho }\left(q\right)=-\frac{\partial \rho }{\partial q^{i}}\frac{\partial }{\partial p_{i}}.$$

Note that the Jacobi–Lie algebra bracket of vector fields of the form $X_{\rho }$ on $T^{*}Q$ vanishes. In the following section, we show that $X_{\rho }$ generates the gauge invariance of the canonical Hamiltonian structures on $T^{*}Q$. For X(Q), GCCL reduces to the complete cotangent lift $X^{c*}$ given in (Equation 21). Moreover, GCCL $X^{c*}$ is the infinitesimal generator of the right action of the diffeomorphism group $Diff\left(Q\right)$ on $T^{*}Q$. Image of $\left(\rho ,X\right)$ in s under the GCCL is(98)$$s\to g:\left(\rho ,X\right)\to \hat{\left(\rho ,X\right)}=X^{c*}+X_{\rho },$$
which is the sum of the vector fields $X^{c*}$ and $X_{\rho }$ in Equation (Equation 97). Notice that GCCL(ρ,X) is a Hamiltonian vector field with the Hamiltonian function, see the work by the authors of [30],(99)$$h_{\left(\rho ,X\right)}\left(p,q\right)=p_{i}X^{i}\left(q\right)+\rho \left(q\right).$$

This result can be seen for instance in the work by the authors of [48]. Moreover, the Lie algebra identity in Equation (Equation 96) gives that(100)$${\left[\hat{\left(\rho_{1},X_{1}\right)},\hat{\left(\rho_{2},X_{2}\right)}\right]}_{g}=\hat{{\left[\left(\rho_{1},X_{1}\right),\left(\rho_{2},X_{2}\right)\right]}_{s}}$$
where the bracket on the left hand side is minus the Jacobi–Lie bracket of vector fields on $T^{*}Q$, whereas the bracket on the right hand side is the semidirect product structure on s given in Equation (Equation 78). The space $\hat{s}$ is a subalgebra of g. The product vector field $\hat{{\left[\left(\rho_{1},X_{1}\right),\left(\rho_{2},X_{2}\right)\right]}_{s}}$ is a Hamiltonian vector field on $T^{*}Q$ with Hamiltonian function(101)$$h_{{\left[\left(\rho_{1},X_{1}\right),\left(\rho_{2},X_{2}\right)\right]}_{s}}=\left\{h_{\left(\rho_{1},X_{1}\right)},h_{\left(\rho_{2},X_{2}\right)}\right\}.$$

We can regard Equation (Equation 99) as a mapping,(102)$$s\to F_{s}\left(T^{*}Q\right)\subset F\left(T^{*}Q\right):\left(X,\rho \right)\to h_{\left(X,\rho \right)},$$
and, due to the identity in Equation (Equation 101), it is an embedding of the algebra s into $F\left(T^{*}Q\right)$. In other words, $F_{s}\left(T^{*}Q\right)$ is an isomorphic copy of s in the space $F\left(T^{*}Q\right)$.

### 3.3. The Dual Spaces and Poisson Brackets

#### 3.3.1. The Dual of TQ and Kuperschmidt–Manin Bracket

The dual $T^{*}Q$ of TQ is the direct sum ${⊕}_{n=0}^{\infty }T_{n}^{*}Q$ of symmetric covariant tensor fields $T_{n}^{*}Q$ of all order [47,49]. In coordinates $\left(q^{i}\right)$, an element of $T^{*}Q$ is given by$$A={⊕}_{n=0}^{\infty }A_{n}={⊕}_{n=0}^{\infty }A_{i_{1}i_{2}\ldots i_{n}}\left(q\right)dq^{i_{1}}⊗\ldots ⊗dq^{i_{n}},$$
where $A_{n}\in T_{n}Q$ is a symmetric covariant tensor field of order n. The pairing between $T^{*}Q$ and TQ is given by the infinite sum(103)$$\langle A,X\rangle ={⊕}_{n=0}^{\infty }\langle A_{n},X_{n}\rangle ={⊕}_{n=0}^{\infty }\int_{Q}A_{i_{1}i_{2}\ldots i_{n}}\left(q\right)X^{i_{1}i_{2}\ldots i_{n}}\left(q\right)d^{3}q,$$
where $d^{3}q$ is a volume form on Q. As the dual of a Lie algebra, $T^{*}Q$ has a Lie–Poisson structure(104)$${\left\{F,H\right\}}_{KM}=-\int_{Q}\langle A\left(q\right),{\left[\frac{\delta F}{\delta A},\frac{\delta H}{\delta A}\right]}_{SC}\left(q\right)\rangle d^{3}q$$
called the Kuperschmidt–Manin bracket. Note that since the result of differentiation is a multivector field, the Schouten concomitant is needed. Here, F and H are functionals on $T^{*}Q$, and the reflexivity assumption takes the particular form δF/δA∈TQ. The bracket inside the integral is the Schouten concomitant, and the pairing inside the integral defined in relation (Equation 103) [45].

#### 3.3.2. The Dual of s and Compressible Fluid Bracket

On the dual space $s^{*}=Den\left(Q\right)\times X^{*}\left(Q\right)$ consisting of densities $Den\left(Q\right)$ and one-form densities $X^{*}\left(Q\right)$, the Lie–Poisson structure in Equation (Equation 2) takes the particular form(105)$$\begin{array}{ccc} {\left\{H,F\right\}}_{CF}\left(\rho ,M\right) & = & -\int_{Q}\langle M,[\frac{\delta H}{\delta M},\frac{\delta F}{\delta M}]\rangle d^{3}q\\ & & -\int_{Q}\rho \left(L_{\frac{\delta H}{\delta M}}\left(\frac{\delta F}{\delta \rho }\right)-L_{\frac{\delta F}{\delta M}}\left(\frac{\delta H}{\delta \rho }\right)\right)d^{3}q, \end{array}$$
where CF stands for compressible fluids.

Ref. [69,70,71]. H and F are two functionals on $s^{*}$ and reflexivity condition is assumed, that is, $\delta F/\delta M\in X\left(Q\right)$ and $\delta F/\delta \rho \in F\left(Q\right)$. To obtain the equations governing the dynamics of isentropic compressible fluid, we choose the Hamiltonian functional(106)$$H\left(\rho ,M\right)=\frac{1}{2}\int_{Q}\frac{M^{2}}{\rho }d^{3}q+\int_{Q}\rho w\left(\rho \right)d^{3}q,$$
which is the total energy of the continuum consisting of a kinetic term and a potential term with internal energy $w=w\left(\rho \right)$.

#### 3.3.3. The Dual of g and Momentum-Vlasov Bracket

The Lie–Poisson bracket on the dual space is given by(107)$${\left\{H,F\right\}}_{mV}\left(\Pi_{f}\right)=-\int_{T^{*}Q}\Pi_{f}\left(z\right)\cdot \left[\frac{\delta H}{\delta \Pi_{f}},\frac{\delta F}{\delta \Pi_{f}}\right]d\mu$$
where the bracket $\left[•,•\right]$ inside the integral is the Jacobi–Lie bracket and dμ is the symplectic volume.

### 3.4. Lifts of Actions to Cotangent Bundle

The cotangent lifts(108)$$\phi_{g}^{R,L}=T^{*}\psi_{g^{-1}}^{R,L}$$
of the left and right actions of $Diff\left(Q\right)$ on Q are the right and left actions on the cotangent bundle $T^{*}Q$, respectively. Both actions, $\phi_{g}^{R,L}$, are canonical, which means they respect canonical symplectic two-form $\Omega_{T^{*}Q}$ on $T^{*}Q$ [51]. Thus, for all g∈G, transformations $\phi_{g}^{R,L}$ are elements of group $Diff_{can}\left(T^{*}Q\right)$ of canonical diffeomorphims on $T^{*}Q$. Infinitesimal generators of the actions $\phi_{g}^{R,L}$ are vector fields on $T^{*}Q$ and computed by(109)$$\frac{d}{dt}{\left.\phi_{z}^{R,L}g\left(t\right)\right|}_{t=0}=T_{e}\phi_{z}^{R,L}\circ X^{R,L}={\left(X^{R,L}\right)}^{c*}$$
where $X^{R,L}$ are vector fields generating $\psi_{g}^{R,L}$. The mappings $\phi_{z}^{R,L}$ are obtained by fixing a point $z\in T^{*}Q$ and they are from $Diff\left(Q\right)$ to $T^{*}Q$. ${\left(X^{R,L}\right)}^{c*}$ are complete cotangent lifts of $X^{R,L}$, as described in Equation (Equation 21).

We define an action *t* of additive group F(Q) to cotangent bundle $T^{*}Q$ by momentum translations. Explicitly, action of ϕ∈F(Q) to an element $z\in T^{*}Q$ over the point $q=\pi_{Q}\left(z\right)$ is(110)$$t:F\left(Q\right)\times T^{*}Q\to T^{*}Q:\left(\phi ,z\right)\to z-d\phi \left(q\right)$$

In a Darboux’ chart, an element *z* of $T^{*}Q$ is represented by $\left(q,p\right)$ and $t_{\phi }\left(q,p\right)=\left(q,p-\nabla_{q}\phi \right)$. The canonical symplectic structure $\Omega_{Q}$ is invariant under the momentum translations, which is the gauge symmetry of Hamiltonian dynamics. In other words, the transformation $t_{\phi }$ is canonical, hence $t_{\phi }\;$ is an element of $Diff_{can}\left(T^{*}Q\right)$. Infinitesimal generator $X_{\phi }\left(q,p\right)=-\nabla_{q}\phi \cdot \nabla_{p}$ is a Hamiltonian vector on $T^{*}Q$ with the Hamiltonian function $\phi =\phi \left(q\right)$ regarded as an element of $F(T^{*}Q)$. The mapping $F\left(Q\right)\to X_{ham}\left(T^{*}Q\right):\rho \to X_{\rho }$ is a Lie algebra isomorphism into in Equation (Equation 97). The following lemma shows that $\phi_{g}^{R,L}$ and $t_{\rho }$ commute up to the actions $\sigma_{g}^{R,L}$ of $Diff\left(Q\right)$ on ρ [34].

**Lemma** **7.**
*The composition of the actions $\phi_{g}^{R,L}$ in Equation (Equation 108) and $t_{\phi }$ in Equation (Equation 110) on $T^{*}Q$ are intertwining, that is,*
(111)$$\phi_{g}^{R,L}\circ t_{\phi }\circ \phi_{g^{-1}}^{R,L}=t_{\sigma_{g}^{R,L}\left(\phi \right)},$$
*where $\sigma_{g}^{R,L}$ are the actions of $Diff\left(Q\right)$ on F(Q) given in Equation (Equation 87).*


**Proof.** Let us consider $z\in T^{*}Q$ over the point $q=\pi_{Q}\left(z\right)$. Then, we have$$\begin{array}{ccc} \phi_{g}^{R,L}\circ t_{\phi }\circ \phi_{g^{-1}}^{R,L}\left(z\right) & = & \phi_{g}^{R,L}\circ t_{\phi }\circ T^{*}\psi_{g}^{R,L}\left(z\right)\\ & = & \phi_{g}^{R,L}\left(T^{*}\psi_{g}^{R,L}\left(z\right)-d\phi \left(\psi_{g^{-1}}^{R,L}\left(q\right)\right)\right)\\ & = & T^{*}\psi_{g^{-1}}^{R,L}\circ \left(T^{*}\psi_{g}^{R,L}\left(z\right)-d\phi \left(\psi_{g^{-1}}^{R,L}\left(q\right)\right)\right)\\ & = & z-d\left({\left(\psi_{g^{-1}}^{R,L}\right)}^{*}\phi \right)=t_{\sigma_{g}^{R,L}\left(\phi \right)}\left(z\right). \end{array}$$ □

This lemma enables us to define two possible embeddings of the semidirect product group *S* into the group of canonical diffeomorphisms *G*, given by(112)$$S↪G:\left(g,\phi \right)\to t_{\phi }\circ \phi_{g}^{R,L},$$
where the actions *t* and ϕ are in Equation (Equation 110) and Equation (Equation 109), respectively. On the Lie algebra level, this turns out be the mapping(113)$$s↪g:\left(X,\phi \right)\to \mathrm{GCCL}\left(X,\phi \right)=X^{c*}+X_{\phi },$$
which is the one in Equation (Equation 98). The infinitesimal version of this lemma is(114)$$\left[{\left(X^{R,L}\right)}^{c*},X_{\phi }\right]=X_{L_{X^{R,L}}\phi }$$
where $X_{L_{\left(X^{R,L}\right)}\phi }$ are Hamiltonian vector fields for the functions obtained by the Lie derivations of ϕ in the directions of $X^{R,L}$. What we derive in Equation (Equation 114) is a particular case of the algebra in Equation (Equation 78).

### 3.5. The Dual Mapping of GCCL

In the previous subsection we showed that the GCCL, mapping TQ to g, is a Lie algebra homomorphism, see Equation (Equation 96). Therefore, its dual mapping $\Phi :g^{*}\to T^{*}Q$ is a momentum and a Poisson mapping [30]. Taking explicitly $\Pi_{f}=\Pi_{i}dq^{i}+\Pi^{i}dp_{i}\in g^{*}$, the dual operation becomes(115)$$\Phi \left(\Pi_{f}\right)={⊕}_{n=0}^{\infty }\int_{T_{q}^{*}Q}\left(\theta_{T^{*}Q}^{n-1}⊗\vartheta \right)d^{3}p,$$
where $\theta_{T^{*}Q}^{n-1}$ is the $\left(n-1\right)$-th tensor power of the canonical one form $\theta_{T^{*}Q}$ and ϑ is a one-form on $T^{*}Q$, given explicitly by$$\vartheta =\left(n\Pi_{i}+\frac{\partial \Pi^{j}}{\partial q^{j}}p_{i}\right)dq^{i}.$$
The definition stems from the duality(116)$$\langle X_{n},\Phi \left(Y^{*}\right)\rangle =\langle X_{n}^{c*},\Pi_{f}\rangle .$$
Left hand side of this equation is the n-th component of the image of Φ while the right hand side can be explicitly calculated from the definition of GCCL.

The image of $\Pi_{f}$ under the dual mapping Φ gives the moments of the momentum-Vlasov dynamics. The n−th moment of $\Pi_{f}$ is given by$$A_{n}=\int_{T_{q}^{*}Q}\left(\theta_{T^{*}Q}^{n-1}⊗\vartheta \right)d^{3}p.$$

Note that the substitution of the momentum map $\Pi_{f}\to f$ in Equation (Equation 52), we have the kinetic moments of the Vlasov dynamics [45,47]. Indeed, the n-th moment reads explicitly(117)$$\begin{array}{ccc} A_{n} & = & \int dpp_{i_{1}}\ldots p_{i_{n-1}}\left(n\Pi_{i_{n}}+p_{i_{n}}\frac{\partial \Pi^{j}}{\partial q^{j}}\right)dq^{i_{1}}\ldots dq^{i_{n}}\\ & = & -\int dpp_{i_{1}}\ldots p_{i_{n}}\left(\frac{\partial \Pi_{j}}{\partial p_{j}}-\frac{\partial \Pi^{j}}{\partial q^{j}}\right)dq^{i_{1}}\ldots dq^{i_{n}}\\ & = & \int dpp_{i_{1}}\ldots p_{i_{n}}f\left(t,q,p\right)dq^{i_{1}}\ldots dq^{i_{n}}, \end{array}$$
where the Ω-divergence is interpreted as the one-particle distribution function $f\left(q,p\right)=\partial \Pi^{j}/\partial q^{j}-\partial \Pi_{j}/\partial p_{j}$. The $A_{n}$ moment is thus the standard n-th moment in kinetic theory (up to some geometrical prefactors dq). In particular, the zero-th moment reads $A_{0}=\int dp\partial \Pi^{i}/\partial q^{i}$, whereas the first moment is $A_{1}=\int dp\left(\Pi_{i}+p_{i}\partial \Pi^{j}/\partial q^{j}\right)dq^{i}$. The following proposition summarizes the situation.

**Proposition** **5.**
*The kinetic moments in Equation (Equation 115) of momentum-Vlasov equations are Poisson mappings from momentum-Vlasov bracket on $g^{*}$ in Equation (Equation 107) to Kuperschmidt–Manin bracket on $T^{*}Q$ given in Equation (Equation 104). In other words,*
(118)$$\Phi^{*}{\left\{F,H\right\}}_{KM}={\left\{\Phi^{*}F,\Phi^{*}H\right\}}_{mV},$$
*holds for all functionals F and H on $T^{*}Q$, see also the work by the authors of [48].*


**Proof.** To prove this fact, we consider a linear functional $F_{X}$ on $T^{*}Q$ in form$$F_{X}\left(A\right)=\langle A,X\rangle ={⊕}_{n=0}^{\infty }\langle A_{n},X_{n}\rangle ={⊕}_{n=0}^{\infty }\int_{Q}A_{i_{1}i_{2}\ldots i_{n}}\left(q\right)X^{i_{1}i_{2}\ldots i_{n}}\left(q\right)d^{3}q.$$Due to linearity, we have that $\delta F_{X}/\delta A=X$. The pull-back $\Phi^{*}F_{X}$ of $F_{X}$ to $g^{*}$ via the momentum map Φ in Equation (Equation 115) is$$\left(\Phi^{*}F_{X}\right)\left(\Pi_{f}\right)={⊕}_{n=0}^{\infty }\int_{T^{*}Q}np_{i_{1}}p_{i_{2}}\ldots p_{i_{n-1}}\left(\Pi_{i_{n}}+p_{i_{n}}\frac{\partial \Pi^{l}}{\partial q^{l}}\right)X^{i_{1}i_{2}\ldots i_{n}}\left(q\right)\Omega_{Q}^{3},$$
where $\Omega_{Q}^{3}$ is the symplectic volume form on the cotangent bundle $T^{*}Q$. Variation of $\Phi^{*}F_{X}$ with respect to its argument $\Pi_{f}$ is$$\frac{\delta \left(\Phi^{*}F_{X}\right)}{\delta \Pi_{f}}=X_{h_{X}}=X^{c*},$$
where $X_{h_{X}}$ is the Hamiltonian vector field corresponding to the Hamiltonian function $h_{X}$ in Equation (Equation 92). The momentum-Vlasov bracket is(119)$$\begin{array}{cc} {\left\{\Phi^{*}F_{X},\Phi^{*}F_{Y}\right\}}_{mV}\left(\Pi_{f}\right)\; & =-\int_{T^{*}Q}\langle \Pi_{f}\left(z\right),\left[\frac{\delta \Phi^{*}F_{X}}{\delta \Pi_{f}},\frac{\delta \Phi^{*}F_{Y}}{\delta \Pi_{f}}\right]\left(z\right)\rangle \Omega_{Q}^{3}\\ \; & =-\int_{T^{*}Q}\langle \Pi_{f}X_{\left\{h_{X},h_{Y}\right\}}\left(z\right)\rangle \Omega_{Q}^{3}\\ \; & =\int_{T^{*}Q}\langle f\left\{h_{X},h_{Y}\right\}\left(z\right)\rangle \Omega_{Q}^{3} \end{array}$$
on $g^{*}$, where the bracket inside the integral is minus the Jacobi–Lie bracket of vector fields satisfying$$\left[\frac{\delta \left(\Phi^{*}F_{X}\right)}{\delta \Pi_{f}},\frac{\delta \left(\Phi^{*}F_{Y}\right)}{\delta \Pi_{f}}\right]=X_{\left\{h_{X},h_{Y}\right\}}.$$Hence, the Poisson map relation in Equation (Equation 118) follows the direct substitutions. □

#### M-Vlasov to Fluid Map

We have established that semidirect product $s^{R}$ is a subalgebra of the space TQ of symmetric contravariant tensor fields. It was also shown that the generalized complete cotangent lift in Equation (Equation 93) reduces to injective homomorphism $s\to \hat{s}\subset g$ in Equation (Equation 98) when restricted to the subalgebra $s^{R}.$ The dual$$\Phi :g^{*}\to s^{*}:\Pi_{f}\to \left(\rho ,M\right)$$
of this Lie algebra homomorphism is the first two moments of momentum-Vlasov dynamics given in Equation (Equation 115). In Darboux’ chart where $\Pi_{f}=\Pi_{i}dq^{i}+\Pi^{i}dp_{i}$, the momentum mapping Φ is explicitly given by(120)$$\rho \left(q\right)=\int_{T_{q}^{*}Q}\frac{\partial \Pi^{i}}{\partial q^{i}}d^{3}p,\;\;\;M_{i}=\int_{T_{q}^{*}Q}\left(\Pi_{i}+p_{i}\frac{\partial \Pi^{j}}{\partial q^{j}}\right)d^{3}p$$
where ρ is a real valued function on Q and $M=M_{i}\left(q\right)dq^{i}$ is a differential one-form on Q, which are the zero-th and first moments $A_{0}$ and $A_{1}$, respectively. Hence, we arrive the following lemma.

**Lemma** **8.**
*The system of mappings in Equation (Equation 120) is a momentum and a Poisson map from the dual $g^{*}$ of Hamiltonian vector fields with momentum-Vlasov bracket in Equation (Equation 119) to the dual $s^{*}$ of the semidirect product space $X\left(Q\right)ⓈF\left(Q\right)$ with compressible fluid bracket in Equation (Equation 105).*


We call the operation in Equation (Equation 120) as “m-Vlasov to fluid map”. The substitution of $g^{*}\to F\left(T^{*}Q\right):\Pi_{f}\to f\left(q,p\right)$ gives plasma to fluid map in the work by the authors of [72].

**Remark** **3**(TQ represents conjugate variables)**.**
*The function on the manifold Q discussed above can be thought of as the conjugate density in the energetic representation, i.e., $\frac{\delta E}{\delta \rho }$ with ρ being density of matter. The function on Q can be thought of as chemical potential, usually denoted by μ. Similarly, the function can also stand for the conjugate entropy density, $T=\frac{\delta E}{\delta s}$, which is the field of temperature. The vector field above can be thought of as the conjugate momentum density, $v=\frac{\delta E}{\delta M}$, which is the velocity field.*

## 4. Geometric Pathways to Fluid Theories

### 4.1. Momentum Formulation of Compressible Fluid Flow

In this section, we shall show how some of the physical theories fit the geometrization procedure presented in (Section 2). For this end, we start with a generic Hamiltonian vector field,(121)$$X_{h}=\frac{\partial h}{\partial p_{i}}\frac{\partial }{\partial q^{i}}-\frac{\partial h}{\partial q^{i}}\frac{\partial }{\partial p_{i}}\in X\left(T^{*}Q\right).$$
on the canonical symplectic manifold $\left(T^{*}Q,\Omega_{Q}\right)$. Then the complete cotangent lift of $X_{h}$ is a vector field on iterated cotangent bundle $T^{*}T^{*}Q$, which can be computed in the Darboux’ coordinates $\left(q^{i},p_{i};\Pi_{i},\Pi^{i}\right)$ as follows,(122)$$X_{h}^{c*}=X_{h}\left(z\right)+\Pi_{f}^{♯}\left(\frac{\partial h}{\partial q^{i}}\right)\frac{\partial }{\partial \Pi_{i}}+\Pi_{f}^{♯}\left(\frac{\partial h}{\partial p_{i}}\right)\frac{\partial }{\partial \Pi^{i}}\in X\left(T^{*}T^{*}Q\right).$$

We use that $\Pi_{f}^{♯}=\Omega_{Q}^{♯}\left(\Pi_{f}\right)$ is the image of a one-form $\Pi_{f}$ under the musical isomorphism induced by the canonical symplectic two-form $\Omega_{Q}$ on $T^{*}Q$. $\Pi_{f}^{♯}$ is given locally by(123)$$\Pi_{f}^{♯}=\Omega_{T^{*}Q}^{♯}\left(\Pi_{f}\right)=\Pi^{i}\frac{\partial }{\partial q^{i}}-\Pi_{i}\frac{\partial }{\partial p_{i}}.$$

Therefore, the action $\Pi_{f}^{♯}\left(\partial h/\partial q^{i}\right)$ in (Equation 122) is simply the action of the vector field $\Pi_{f}^{♯}$ on the real valued function $\partial h/\partial q^{i}$. It is interesting to note that $X_{h}^{c*}$ is a Hamiltonian vector field on the symplectic manifold $\left(T^{*}T^{*}Q,\Omega_{T^{*}Q}\right)$ with the Hamiltonian function $\langle \Pi ,X_{h}\rangle$, that is,(124)$$i_{X_{h}^{c*}}\Omega_{T^{*}Q}=d\langle X_{h},\Pi_{f}\rangle .$$

The decomposition of the complete cotangent lift $X_{h}^{c*}$ into the sum its vertical representative $VX_{h}^{c*}$ and its holonomic part $HX_{h}^{c*}$ are computed to be(125)$$\begin{array}{ccc} VX_{h}^{c*} & = & \left(\Pi_{f}^{♯}\left(\frac{\partial h}{\partial q^{i}}\right)-X_{h}\left(\Pi_{i}\right)\right)\frac{\partial }{\partial \Pi_{i}}+\left(\Pi_{f}^{♯}\left(\frac{\partial h}{\partial p_{i}}\right)-X_{h}\left(\Pi^{i}\right)\right)\frac{\partial }{\partial \Pi^{i}},\\ HX_{h}^{c*} & = & X_{h}+X_{h}\left(\Pi_{i}\right)\frac{\partial }{\partial \Pi_{i}}+X_{h}\left(\Pi^{i}\right)\frac{\partial }{\partial \Pi^{i}}. \end{array}$$

#### 4.1.1. Momentum-Euler Equations

It was shown in the previous section that the generalized complete cotangent lift determines an embedding $s\to \hat{s}\subset g$, as given in Equation (Equation 98). The image $\hat{\left(X,\phi \right)}$ is a Hamiltonian vector field on $T^{*}Q$. The complete cotangent lift of $\hat{\left(X,\phi \right)}$ is the Hamiltonian vector field(126)$${\left(\hat{\left(X,\phi \right)}\right)}^{c*}=\hat{\left(X,\phi \right)}+\left(\Pi^{k}p_{j}\frac{\partial^{2}X^{j}}{\partial q^{k}\partial q^{i}}+\Pi^{k}\frac{\partial^{2}\phi }{\partial q^{k}\partial q^{i}}-\Pi_{k}\frac{\partial X^{k}}{\partial q^{i}}\right)\frac{\partial }{\partial \Pi_{i}}+\left(\Pi^{k}\frac{\partial X^{i}}{\partial q^{k}}\right)\frac{\partial }{\partial \Pi^{i}}$$
on $T^{*}T^{*}Q$ satisfying the Hamilton’s equations in (Equation 124) with the Hamiltonian function$$\langle \Pi_{f},\hat{\left(X,\phi \right)}\rangle \left(q,p\right)=X^{i}\Pi_{i}-p_{j}\frac{\partial X^{j}}{\partial q^{i}}\Pi^{i}-\frac{\partial \phi }{\partial q^{i}}\Pi^{i}.$$

The vertical representative of the cotangent lift ${\left(\hat{\left(X,\phi \right)}\right)}^{c*}$ is a generalized vector field of order 1 and is given by the following abbreviated formula,(127)$$V{\left(\hat{\left(X,\phi \right)}\right)}^{c*}={\dot{\Pi }}_{i}\frac{\partial }{\partial \Pi_{i}}+{\dot{\Pi }}^{i}\frac{\partial }{\partial \Pi^{i}}$$
where the coefficient functions are locally in the form(128)$$\begin{array}{ccc} {\dot{\Pi }}_{i} & = & \Pi^{k}p_{j}\frac{\partial^{2}X^{j}}{\partial q^{k}\partial q^{i}}+\Pi^{k}\frac{\partial^{2}\phi }{\partial q^{k}\partial q^{i}}-\Pi_{k}\frac{\partial X^{k}}{\partial q^{i}}-X^{k}\frac{\partial \Pi_{i}}{\partial q^{k}}+\left(p_{k}\frac{\partial X^{k}}{\partial q^{j}}+\frac{\partial \phi }{\partial q^{j}}\right)\frac{\partial \Pi_{i}}{\partial p_{j}}\\ {\dot{\Pi }}^{i} & = & \Pi^{k}\frac{\partial X^{i}}{\partial q^{k}}-X^{k}\frac{\partial \Pi^{i}}{\partial q^{k}}+\left(p_{k}\frac{\partial X^{k}}{\partial q^{j}}+\frac{\partial \phi }{\partial q^{j}}\right)\frac{\partial \Pi^{i}}{\partial p_{j}}. \end{array}$$

We call the system of equations given in (Equation 128) the momentum-Euler equations. In the density variable these system of equations reduces to(129)$$\frac{\partial f}{\partial t}+X^{i}\frac{\partial f}{\partial q^{i}}-p_{j}\frac{\partial X^{j}}{\partial q^{i}}\frac{\partial f}{\partial p_{i}}-\frac{\partial \phi }{\partial q^{i}}\frac{\partial f}{\partial p_{i}}=0$$
by the substitution of the momentum map in Equation (Equation 52) into Equation (Equation 128). Note that $X^{i}$ can be though of as the *i*-th component of the fluid velocity, and ϕ can be thought of as the chemical potential. Equation (Equation 129) can be interpreted physically as dynamics of fluctuations around mean velocity field $X^{i}$ and field of chemical potential ϕ.

Geometrization of the right hand side of Equation (Equation 127) can also be described as follows. Vertical lift of the one-form $\Pi_{f}$ is a vector field:(130)$$\Pi_{f}^{v}=\Omega_{T^{*}Q}^{♯}\circ T^{*}\pi_{T^{*}Q}\circ \Pi_{f}\circ \pi_{T^{*}Q}:T^{*}T^{*}Q\to TT^{*}T^{*}Q$$
where $T^{*}\pi_{T^{*}Q}$ is the cotangent lift of the projection $\pi_{T^{*}Q}:T^{*}T^{*}Q\to T^{*}Q$ and $\Omega_{T^{*}Q}^{♯}$ is the musical isomorphism induced from the canonical symplectic form $\Omega_{T^{*}Q}$ on $T^{*}T^{*}Q$. Hence, momentum-Euler equations can be written as$${\dot{\Pi }}_{f}^{v}=V{\left(\hat{\left(X,\phi \right)}\right)}^{c*},$$
where the dot on the left hand side denotes the time derivative.

#### 4.1.2. Back to the Classical Form of the Compressible Fluid

To turn back to the familiar formulation of Euler’s equation, we first substitute the m-Vlasov to fluid map in Equation (Equation 120) into the m-Euler Equations (Equation 128). This gives an intermediate class of equations(131a)$$\begin{array}{ccc} \dot{M} & = & -L_{X}M-div\left(X\right)M-\rho d\phi \end{array}$$
(131b)$$\begin{array}{ccc} \dot{\rho } & = & -div\left(\rho X\right). \end{array}$$

If we change coordinates to(132)$$X^{i}\rho =\delta^{ij}M_{j}\;\;\;\mathrm{and}\;\;\;\phi =\frac{M^{2}}{\rho^{2}}+h\left(\rho \right)$$
in system ([Disp-formula FD131b-entropy-21-00907]), we obtain the equations for compressible fluids(133)$$\frac{\partial X}{\partial t}+\left(X\cdot \nabla \right)X=\frac{1}{\rho }\nabla p\;\;\;\mathrm{and}\;\;\;\dot{\rho }+div\rho X=0.$$
in standard formulation. The first of this substitution in Equation (Equation 132) is simple relation between velocity and momentum, and the second one is related to Bernoulli’s theorem for isentropic fluid flows. Here, $h\left(\rho \right)=\rho w^{\prime }+w$ is the enthalpy function and $w=w\left(\rho \right)$ is the internal energy of the continuum. Yet another form of Equation (131) is(134a)$$\begin{array}{ccc} \dot{\rho } & = & -\partial_{k}\left(\rho X^{k}\right) \end{array}$$
(134b)$$\begin{array}{ccc} {\dot{M}}_{i} & = & -\rho \partial_{i}\phi -M_{j}\partial_{i}X^{j}-\partial_{j}\left(M_{i}X^{j}\right), \end{array}$$
where ϕ=δE/δρ and $X^{k}=\delta E/\delta m_{k}$ are chemical potential and velocity.

These are the usual equations for fluid mechanics in absence of entropy (or isentropic), see, e.g., the work by the authors of [13]. Entropy density can be added as follows. In kinetic theory entropy density is defined as(135)s(q)=∫dpη(f(q,p)),
where η(f) is a real function of real variable *f*, e.g., $-k_{B}f\left(\ln\left(h^{3}f\right)-1\right)$ for ideal gases and $k_{B}$ and *h* are the Boltzmann and Planck constants, respectively. Evolution of this field is then given by Equation (Equation 129),(136)$$\partial_{t}s=\int dp\eta^{\prime }\left(f\right)\partial_{t}f=-\partial_{k}\left(sX^{k}\right),$$
which is the usual law of entropy conservation. However, to recover the antisymmetric coupling between *s* and $M_{i}$, one should add the term $-s\partial_{i}E_{s}$ to the evolution equation for $M_{i}$. The evolution equations are then completely equivalent to the evolution equations of fluid mechanics coming from the underlying Poisson bracket, e.g., the work by the authors of [13].

### 4.2. The 10-Moment Approximation

In this section, we present a generalization of the momentum-Euler equations to ten kinetic moments (1 density + 3 momentum densities + 6 second moments). The procedure is analogical to construction of the momentum-Euler equations with the only difference that along the ϕ and $X^{i}$ fields there is a $R^{ij}$ tensor field on the base manifold Q.

#### 4.2.1. Double GCCL of the Second Order Tensor Field

Consider a second order symmetric contravariant tensor field on Q given by(137)$$X=\left(\phi \left(q\right),X^{i}\left(q\right)\frac{\partial }{\partial q^{i}},R^{ij}\left(q\right)\frac{\partial }{\partial q^{i}}⊗\frac{\partial }{\partial q^{j}}\right),$$
representing chemical potential, velocity, and the conjugate pressure tensor. Using the mapping (Equation 92), we define the following Hamiltonian function, on $T^{*}Q$,(138)$$h=\phi \left(q\right)+X^{i}\left(q\right)p_{i}+R^{ij}\left(q\right)p_{i}p_{j}.$$

The GCCL of X is then(139)$$\hat{X}=\left(X^{i}+2p_{j}R^{ji}\right)\frac{\partial }{\partial q^{i}}+\left(-\frac{\partial \phi }{\partial q^{i}}-p_{j}\frac{\partial X^{i}}{\partial q^{j}}-p_{j}p_{k}\frac{\partial R^{jk}}{\partial q^{i}}\right)\frac{\partial }{\partial p_{i}},$$
which is a vector field on $T^{*}Q$. The subsequent GCCL of $\hat{X}$ gives(140)$$\begin{array}{ccc} \hat{\hat{X}} & = & \hat{X}-\Pi_{m}\frac{\partial X^{m}+2p_{j}R^{jm}}{\partial q^{i}}\frac{\partial }{\partial \Pi_{i}}-\Pi_{m}\frac{\partial X^{m}+2p_{j}R^{jm}}{\partial p_{i}}\frac{\partial }{\partial \Pi^{i}}\\ & & -\Pi^{m}\frac{\partial -\frac{\partial \phi }{\partial q^{i}}-p_{j}\frac{\partial X^{i}}{\partial q^{j}}-p_{j}p_{k}\frac{\partial R^{jk}}{\partial q^{i}}}{\partial q^{i}}\frac{\partial }{\partial \Pi_{i}}-\Pi_{m}\frac{\partial -\frac{\partial \phi }{\partial q^{i}}-p_{j}\frac{\partial X^{i}}{\partial q^{j}}-p_{j}p_{k}\frac{\partial R^{jk}}{\partial q^{i}}}{\partial p_{i}}\frac{\partial }{\partial \Pi^{i}}, \end{array}$$
which is a vector field on $T^{*}T^{*}Q$.

#### 4.2.2. Vertical Representative

The vertical representative $V(\hat{\hat{X}})$ of the second GCCL of X is(141)$$\begin{array}{c} V\left(\hat{\hat{X}}\right)={\hat{\hat{X}}}_{v}-\hat{\hat{X}}\left(\Pi^{i}\right)\frac{\partial }{\partial \Pi^{i}}-\hat{\hat{X}}\left(\Pi_{i}\right)\frac{\partial }{\partial \Pi_{i}}, \end{array}$$
where ${\hat{\hat{X}}}_{v}=\hat{\hat{X}}-\hat{X}$. This vector field has only components in the directions of $\Pi^{i}$ and $\Pi_{i}$, and the components are then interpreted as evolution equations for $\Pi^{i}$ and $\Pi_{i}$,(142a)$$\begin{array}{ccc} \partial_{t}\Pi^{i} & = & -\Pi_{j}\frac{\partial X^{j}}{\partial q^{i}}-2\Pi_{k}p_{j}\frac{\partial R^{jk}}{\partial q^{i}}+\Pi^{j}\frac{\partial^{2}\phi }{\partial q^{i}\partial q^{j}}+\Pi^{k}p_{j}\frac{\partial^{2}X^{j}}{\partial q^{i}\partial q^{k}}+p_{j}p_{k}\frac{\partial^{2}R^{jk}}{\partial q^{i}\partial q^{l}}\Pi^{l}\\ & & -X^{k}\frac{\partial \Pi_{i}}{\partial q^{k}}-2p_{j}R^{jk}\frac{\partial \Pi_{i}}{\partial q^{k}}+\frac{\partial \phi }{\partial q^{k}}\frac{\partial \Pi_{i}}{\partial p_{k}}+p_{j}\frac{\partial X^{j}}{\partial q^{k}}\frac{\partial \Pi_{i}}{\partial p_{k}}+p_{j}p_{l}\frac{\partial R^{jl}}{\partial q^{k}}\frac{\partial \Pi_{i}}{\partial p_{k}} \end{array}$$
(142b)$$\begin{array}{ccc} \partial_{t}\Pi_{i} & = & -2\Pi_{j}R^{ji}+\Pi^{j}\frac{\partial X^{i}}{\partial q^{j}}+\Pi^{j}\frac{\partial R^{ik}}{\partial q^{j}}p_{k}+\Pi^{k}\frac{\partial R^{ji}}{\partial q^{k}}p_{j}\\ & & -X^{k}\frac{\partial \Pi^{i}}{\partial q^{k}}-2p_{j}R^{jk}\frac{\partial \Pi^{i}}{\partial q^{k}}+\frac{\partial \phi }{\partial q^{k}}\frac{\partial \Pi^{i}}{\partial p_{k}}+p_{j}\frac{\partial X^{j}}{\partial q^{k}}\frac{\partial \Pi^{i}}{\partial p_{k}}+p_{j}p_{l}\frac{\partial R^{jl}}{\partial q^{k}}\frac{\partial \Pi^{i}}{\partial p_{k}}, \end{array}$$
which contain as a special case $R^{ij}=0$ the momentum-Euler equations (Equation 128).

The distribution function $f\left(q,p\right)=\frac{\partial \Pi^{i}}{\partial q^{i}}-\frac{\partial \Pi_{i}}{\partial q_{i}}$ then evolves as(143)$$\begin{array}{ccc} \partial_{t}f & = & -X^{k}\frac{\partial f}{\partial q^{k}}+\left(\frac{\partial \phi }{\partial q^{k}}+p_{j}\frac{\partial X^{j}}{\partial q^{k}}\right)\frac{\partial f}{\partial p_{k}}\\ & & -2p_{j}R^{jk}\frac{\partial f}{\partial q^{k}}+p_{j}p_{l}\frac{\partial R^{jl}}{\partial q^{k}}\frac{\partial f}{\partial p_{k}}. \end{array}$$

This equation can be interpreted physically as kinetic theory of fluctuations around mean fields of velocity, chemical potential and conjugate pressure tensor.

#### 4.2.3. Projection to Moments

Subsequent projection to density, momentum density, and kinetic stress tensor,(144a)$$\begin{array}{ccc} \rho (q) & = & \int dpf(q,p) \end{array}$$
(144b)$$\begin{array}{ccc} M_{i}\left(q\right) & = & \int dpp_{i}f\left(q,p\right) \end{array}$$
(144c)$$\begin{array}{ccc} P_{ij}\left(q\right) & = & \int dpp_{i}p_{j}f\left(q,p\right) \end{array}$$
leads to evolution equations for the projected variables (by chain rule)(145a)$$\begin{array}{ccc} \partial_{t}\rho & = & -\partial_{k}\left(\rho X^{k}\right)-2\partial_{k}\left(R^{jk}m_{j}\right) \end{array}$$
(145b)$$\begin{array}{ccc} \partial_{t}M_{i} & = & -\rho \partial_{i}\phi -M_{j}\partial_{i}X^{j}-P_{jk}\partial_{i}R^{jk}-\partial_{j}\left(M_{i}X^{j}\right)-\partial_{k}\left(R^{kj}P_{ij}\right)-\partial_{k}\left(R^{jk}P_{ij}\right) \end{array}$$
(145c)$$\begin{array}{ccc} \partial_{t}P_{ij} & = & -\partial_{k}\left(P_{ij}X^{k}\right)-M_{j}\partial_{i}\phi -M_{i}\partial_{j}\phi -\partial_{i}X^{k}P_{jk}-\partial_{j}X^{k}P_{ik}\\ & & -2\partial_{k}\left(R^{kl}Q_{ijl}\right)-\partial_{i}R^{kl}Q_{jkl}-\partial_{j}R^{kl}Q_{ikl}, \end{array}$$
where(146)$$Q_{ijk}=\int dpp_{i}p_{j}p_{k}f\left(q,p\right)$$
are the higher higher moments. The evolution equation for the stress tensor is thus not in a closed form, which is typical in the Grad hierarchy, e.g., the work by the authors of [62].

Let us now seek an appropriate closure, i.e., specification of the $Q_{ijk}$ terms. Note first that the fields ϕ, $X^{i}$ and $R^{ij}$ can be interpreted as the corresponding derivatives of energy with respect to the kinetic moments,(147)$$\phi =\frac{\delta E}{\delta \rho },\;X^{i}=\frac{\delta E}{\delta M_{i}}\;\mathrm{and}\;R^{ij}=\frac{\delta E}{\delta P_{ij}}.$$

In other words, ϕ is chemical potential, $X^{i}$ is velocity, and $R^{ij}$ is the conjugate variable to the second kinetic moments. We now seek the closure so that energy is conserved regardless the choice of energy, which requires equations (145) to possess antisymmetric structure. The coupling between the evolution of ρ and evolution of $M_{i}$ is antisymmetric as can be directly verified by construction of the generating Poisson bracket. To make the coupling between $m_{i}$ and $Q_{ijk}$ antisymmetric as well, we have to set $Q_{ijk}=0$, which is the sought closure. Besides automatic energy conservation, Jacobi identity is then satisfied for the evolution equations as shown, for instance, in the work by the authors of [73]. The closure can be thus referred to as a Hamiltonian closure.

#### 4.2.4. Adding Entropy

As in the case of momentum-Euler equations, entropy density can be defined as(148)s(q)=∫dpη(f).
It evolves due to Equation (Equation 143) as(149)$$\partial_{t}s=-\partial_{k}\left(sX^{k}\right)-2R^{jk}\partial_{k}b_{j}-\partial_{j}R^{jl}b_{l}-\partial_{l}R^{jl}b_{j},$$
where(150)$$b_{i}=\int dpp_{i}\eta \left(f\right)$$
is the first entropic moment (considered also in the work by the authors of [62].

Interestingly, the evolution for the first entropic moment is (using again Equation (Equation 143))(151)$$\begin{array}{ccc} \partial_{t}b_{i} & = & -\partial_{j}\left(b_{i}X^{j}\right)-s\partial_{i}\phi -b_{j}\partial_{i}v^{j}\\ & & -2R^{jk}\partial_{k}B_{ij}-B_{jl}\partial_{i}R^{jl}-B_{il}\partial_{j}R^{jl}-B_{ij}\partial_{l}R^{jl}, \end{array}$$
where(152)$$B_{ij}=\int dpp_{i}p_{j}\eta \left(f\right)$$
is the tensor of second entropic moments. This way a hierarchy of entropic moments coupled to the hierarchy of kinetic moments can be constructed (similar to kinetic theory of non-ideal gases [74]).

The coupling to the kinetic moments is made antisymmetric by adding complementary terms (in the case of *s* among the state variables) to the equations for $M_{i}$ and $P_{ij}$.

#### 4.2.5. Central Kinetic Moments

In the evolution equations for the kinetic moments (145), it is interesting that the density evolves not solely due to the advection by velocity $X^{i}$. This might appear strange at first sight, but it is actually due to the definition of $P_{ij}$ as the second kinetic moments, not the central second kinetic moments,(153)$$\begin{array}{ccc} {\hat{P}}_{ij} & = & \int dp\left(p_{i}-M_{i}/\rho \right)\left(p_{j}-M_{j}/\rho \right)f\\ & = & P_{ij}-\frac{M_{i}M_{j}}{\rho }. \end{array}$$

The transformation of variables from (ρ,M,P) to $\left(\rho ,M,\hat{P}\right)$ turns derivatives of energy to(154a)$$\begin{array}{ccc} \frac{\delta E}{\delta P_{ij}} & = & \frac{\delta E}{\delta {\hat{P}}_{ij}} \end{array}$$
(154b)$$\begin{array}{ccc} {\left(\frac{\delta E}{\delta M_{i}}\right)}_{P} & = & {\left(\frac{\delta E}{\delta M_{i}}\right)}_{\hat{P}}-\frac{m_{l}}{\rho }\left(\frac{\delta E}{\delta R_{il}}+\frac{\delta E}{\delta R_{li}}\right) \end{array}$$
(154c)$$\begin{array}{ccc} {\left(\frac{\delta E}{\delta \rho }\right)}_{P} & = & {\left(\frac{\delta E}{\delta \rho }\right)}_{\hat{P}}+\frac{\delta E}{\delta R_{ij}}\frac{M_{i}M_{j}}{\rho^{2}}. \end{array}$$

The evolution equation for density ([Disp-formula FD145a-entropy-21-00907]) then becomes(155)$$\partial_{t}\rho =-\partial_{k}\left(\rho {\left(\frac{\delta E}{\delta M_{k}}\right)}_{\hat{P}}\right),$$
which already has the usual form.

## 5. Discussion and Conclusions

An injective Lie algebra homomorphism called generalized complete cotangent lift (GCCL) was defined in Equation (Equation 95), mapping the space TQ of generalized symmetric contravariant tensor fields on a manifold Q to the space $g=X_{ham}\left(T^{*}Q\right)$ of Hamiltonian vector fields on the cotangent bundle $T^{*}Q$. It has been shown that kinetic moments in Equation (Equation 115) of the momentum-Vlasov equations represent Poisson mappings obtained by the dualization of this homomorphism. The configuration space of compressible isentropic fluids is the semidirect product group $S=Diff\left(Q\right)ⓈF\left(Q\right)$, and its Lie algebra, $s=X\left(Q\right)ⓈF\left(Q\right)$, is a subalgebra of TQ. Restriction of GCCL on s gives that embedding s↪g, whereas the intertwining lemma and Equation (Equation 111) establish this embedding on the group level, that is, $S↪G=Diff_{can}\left(T^{*}Q\right)$. The dual of the embedding s↪g, called *m-Vlasov to fluid map* in Equation (Equation 120), is a Poisson and momentum mapping relating m-Vlasov bracket in Equation (Equation 107) and compressible fluid bracket in Equation (Equation 105). Generalized complete cotangent lift $\hat{\left(X,\phi \right)}$ of a pair $\left(X,\phi \right)\in s$ is a Hamiltonian vector field on $T^{*}Q$. We have introduced momentum-Euler equations,(156)$$V{\left(\hat{\left(X,\rho \right)}\right)}^{c*}=\Pi^{v},$$
by taking the vertical representative of the complete lift of $\hat{\left(X,\rho \right)}$. It is shown that after the substitution of “m-Vlasov to fluid map” and the coordinate transformation in Equation (Equation 132), momentum-Euler equations reduce to compressible fluid equation in the classical form. Thus, it is achieved to arrive Euler’s equation starting from the particle motion and applying pure geometric operations. Note that this geometrization procedure does not need any Hamiltonian functional or Poisson structure. Finally, the reversible evolution equations for ten kinetic moments are found by the same procedure, and a hierarchy of entropic moments coupled to the kinetic moments is identified including entropy. The approach to mechanics presented in this paper can lead to another geometrization of the GENERIC framework [13,17], as it does not rely on Poisson brackets although being purely geometric.

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
