# Peer review of "Lifts of Symmetric Tensors: Fluids, Plasma, and Grad Hierarchy"

_entropy, 2019, doi:10.3390/e21090907_

Round 1

Reviewer 1 Report

The paper in itself is fine. In my opinion, the results should be better delineated to those in the literature, in particular in Mechanics and Symmetry by Marsden and Ratiu and the work by Cesare Tronci and Darryl Holm in a series of papers (some of them are also not cited at all, e.g. D. D. Holm and C. Tronci, “Geodesic Vlasov equations and their integrable moment closures,” J. Geom. Mech., vol. 1, no. 2, pp. 181–208, 2009). It should be stated clearly which results were known in the literature and which ones are new, and also what benefits / additional insights are provided by the results in the paper.

Also, while the English language is overall fine, it can be improved in places (see the attached PDF for some suggestions)

Author Response

Dear Reviewer,

we would firstly like to thank you for your time when reading our manuscript. You have made many valuable comments, improving the manuscript.

We have tried to address all the issues indicated in the pdf file (see the highlighted differences).

Moreover, in a new paragraph beginning on line 83 (as well as on another places) we indicate that a lot of the results have already been obtained in the works by D. D. Holm et al. (see the references for the additional citations), but that there is also an important difference between the approach we are presenting in this paper and the classical Lie-Poisson setting.

In the present work, we are deriving the Lie-Poisson equations using the geometric pathway (complete lifts and vertical representatives) in Section (2). This approach is free from Poisson brackets and Hamiltonian functionals and provides an alternative way towards the results, both geometric and physical (e.g. Grad hierarchy).

We hope that these amendments make the manuscript clearer.

Reviewer 2 Report

The author examined the geometrical and algebraic aspects of the Hamiltonian realizations of the Euler’s fluid and the Vlasov’s plasma-based on efficient theories. The manuscript is well written and is in the interest of the fluid mechanic's communities.

The following are the major comments:

1-Fix some typos and improve the manuscript.

2-Remove the contents section.

3-Improve the introduction section more by discussing clearly the following Refs.:

(1) https://doi.org/10.1007/s40819-018-0513-y

(2) http://dx.doi.org/10.5098/hmt.12.3 

(3) https://doi.org/10.1515/jnet-2018-0099

(4) https://doi.org/10.1016/j.aej.2015.09.015

(5) https://doi.org/10.1016/j.molliq.2016.03.046

(6) https://doi.org/10.1016/j.molliq.2016.04.014

4- Enrich your interesting results by adding some applications of the present outcomes illustrated by tabular and graphical representations.

5-Add nomenclature with SI units.

Author Response

Dear Reviewer,

thank you for your support and suggestions for improvements. In the new version we have fixed the typos and English and we have removed the Contents. Regarding the references you mentioned, we are now briefly discussing a few of them although we do not see much intersection with the manuscript.
We have also included a diagram you asked for elucidating the geometrical construction (Page 6), and physical interpretations of various results have been added (see the highlighted differences). Finally, we have added the nomenclature for terms where considered it relevant.

We hope that these amendments make the manuscript clearer.

Round 2

Reviewer 2 Report

1-  Rearrange the citations increasingly using Mendeley.

2- Correct Ref. 76 as:

Wakif, M. Qasim, M.I. Afridi, S. Saleem, M.M. Al-Qarni, Numerical Examination of the Entropic Energy Harvesting in a Magnetohydrodynamic Dissipative Flow of Stokes’ Second Problem: Utilization of the Gear-Generalized Differential Quadrature Method, Journal of Non-Equilibrium Thermodynamics. (2019) 1–19. doi:10.1515/jnet-2018-0099.

3-Enrich your work physically by discussing the following Refs. in the introduction part:

https://doi.org/10.1140/epjp/i2018-12037-7

https://doi.org/10.1016/j.molliq.2016.03.046

https://doi.org/10.1016/j.molliq.2016.04.014

Author Response

We thank the Reviewer for her/his suggestions. The papers that she/he
suggests to include in citations are interesting, but they are not
related to the subject of our manuscript. We have, however, corrected the mentioned reference.